# Infusing structural assumptions into dimensionality reduction for single-cell RNA sequencing data to identify small gene sets

Maren Hackenberg [1,2,5] ✉, Niklas Brunn [1,2,5] ✉, Tanja Vogel [3] & Harald Binder[1,2,4]

Dimensionality reduction greatly facilitates the exploration of cellular heterogeneity in single-cell RNA sequencing data. While most of such approaches are data-driven, it can be useful to incorporate biologically plausible assumptions about the underlying structure or the experimental design. We propose the boosting autoencoder (BAE) approach, which combines the advantages of unsupervised deep learning for dimensionality reduction and boosting for formalizing assumptions. Specifically, our approach selects small sets of genes that explain latent dimensions. As illustrative applications, we explore the diversity of neural cell identities and temporal patterns of embryonic development.

Dimensionality reduction approaches are an essential component of exploratory single-cell RNA sequencing (scRNA-seq) data analysis to identify underlying patterns of cellular heterogeneity, e.g., by providing visualizations in two dimensions[1,2]. Such low-dimensional representations are then often used for further downstream analyses to answer more targeted questions including cell clustering and trajectory inference[3–5]. However, it might be beneficial to already encode additional assumptions directly into the dimensionality reduction, to find a representation that is specifically tailored to structural knowledge or intuition about a dataset, e.g., corresponding to an intuition about expected patterns or knowledge about experimental design. For example, researchers might expect groups of cells to be captured in different dimensions, which are each characterized by a distinct, small set of genes. When collecting data at different time points during a developmental process, researchers may seek to identify sets of genes that explain the heterogeneity at each time point, but in particular also changes between time points. This requires a flexible approach that can be adapted to incorporate different types of assumptions, such as about experimental design or expected structure. To achieve this, we propose to combine dimensionality reduction with variable selection, where assumptions are formulated as constraints. For example, a modeler could decide that only genes that are complementary to the information encoded in the other dimensions should be selected for a particular reduced dimension. This would ensure that different dimensions capture distinct factors of variation, corresponding to different cell groups, to provide a targeted understanding of cellular heterogeneity.

Currently, nonparametric approaches such as t-SNE[6,7] and UMAP[8,9] are the most widely used to perform dimensionality reduction of omics data for visualization, due to the resulting visually appealing, fine-grained two-dimensional representations. As they are optimized in a fully unsupervised, algorithmic manner, it is not straightforward to encode additional constraints and link specific patterns in the low-dimensional space to individual explanatory genes. While this could in principle be addressed in corresponding parametric alternatives (e.g., refs. [10–15]), where a mapping to the latent space is learned as an explicit function specified by a neural network, it may then be more attractive to directly use neural network-based approaches for dimensionality reduction.

Specifically, autoencoders and variational autoencoders[16] have been used to learn neural network-based low-dimensional latent representations of single-cell transcriptomics (e.g., refs. [10,17–19]), multi-omics[20,21], or spatial transcriptomics data[22]. Autoencoder-based models have also been adapted to encode additional structure in a latent representation, e.g., using external data in single-cell applications[23–25], or to specifically provide sparse[26] or disentangled latent representations[27–29]. However, such approaches are typically designed to satisfy a single specific constraint, and cannot be flexibly adapted to encode different types of structure in a straightforward manner[30]. Since their optimization often involves several additional penalty terms that must be carefully calibrated in the overall loss function, such models tend to be quite complex even for a single constraint. Another drawback of deep neural network models is their natural lack of interpretability due to architectures that typically include multiple nonlinear

[1]Institute of Medical Biometry and Statistics (IMBI), Faculty of Medicine and Medical Center, University of Freiburg, Freiburg, Germany. [2]Freiburg Center for Data Analysis, Modeling and AI, University of Freiburg, Freiburg, Germany. [3]Institute of Anatomy and Cell Biology, Department Molecular Embryology, Faculty of Medicine, University of Freiburg, Freiburg, Germany. [4]Centre for Integrative Biological Signaling Studies (CIBSS), University of Freiburg, Freiburg, Germany. [5]These authors contributed equally: Maren Hackenberg, Niklas Brunn. ✉e-mail: maren.hackenberg@uniklinik-freiburg.de; niklas.brunn@uniklinik-freiburg.de

layers. While multiple layers for dimensionality reduction allow for more flexibility in the learning task, which can help to construct a well-structured embedding of the data in a latent space, it is particularly difficult to determine the most explanatory genes for the learned patterns in different latent dimensions. Therefore, neural network-based approaches often rely on post-hoc variable attribution to link groups of genes to specific latent dimensions[31–33], where an additional analysis step is applied to an already trained model (e.g., refs. 34,35). Still, there are some approaches that incorporate interpretability already as part of the model design. For example, the siVAE approach simultaneously infers a cell and gene embedding space via two encoder-decoder frameworks and and uses an additional regularization term in the loss function, where embeddings of genes indicate their contribution to distinct dimensions of the cell embedding[36]. However, the dimensions in the cell embedding may still be entangled, and the contribution of variables to the dimensions of the cell embedding is not constrained to be sparse.

Similarly, simpler dimensionality reduction approaches, such as principal component analysis (PCA), allow direct inspection of the contribution of individual variables to any latent dimension[37,38], but again provide a dense representation instead of distinct, small sets of genes associated with each dimension. Approaches based on $L_1$ penalties have been proposed to enforce sparsity, i.e., to obtain selection of gene sets[39,40], but these do not provide the dimensionality reduction flexibility of neural networks.

While neural networks can in principle be combined with $L_1$ penalties (e.g., refs. 41,42), we propose to use a more flexible approach for variable selection, namely componentwise boosting[43,44], which has been successfully applied in various omics applications (e.g., refs. 45–48). This approach provides variable selection and regularization similar to $L_1$ penalties[49]. However, it relies on separate variable selection and parameter update components. The selection component can be flexibly adapted (e.g., ref. 50). When combined with neural networks for dimensionality reduction, this could allow to incorporate structural assumptions, e.g., such as each latent pattern being associated with a small set of explanatory genes, or such as the presence of gradually evolving differentiation trajectories, or distinct groups of cells.

Yet, end-to-end models with componentwise boosting and neural networks are not available so far. The challenge in this is that parameters in autoencoder neural network architectures are trained by minimizing a reconstruction loss via stochastic gradient descent in parameter space, while boosting approaches rely on gradients in function space[51]. To overcome this challenge, we propose the boosting autoencoder (BAE) approach. Specifically, the gradient of the reconstruction loss, computed via backpropagation through the decoder component of an autoencoder, serves as a feedback signal for then building the encoder part via componentwise boosting. Joint optimization in an end-to-end model is facilitated by differentiable programming, a paradigm in which gradients of a joint loss function for simultaneous optimization of potentially diverse model components are obtained via automatic differentiation[52,53].

Due to the sparsity constraint, the proposed BAE approach should allow to find small gene sets that characterize very small cell groups with a distinct transcriptomic signature, which might be lost in a global clustering[54]. In contrast to the end-to-end approach we are aiming for, standard analyses to characterize different cell groups by marker genes typically correspond to two-step approaches, where clustering and subsequent analysis by post-hoc differential gene expression tests are performed separately[55,56]. Even if a very small group is captured by such a clustering, reliably inferring marker genes by differential expression testing is often challenging due to the limited sample size[57,58]. Dedicated approaches have been developed to address this problem (e.g., refs. 58–60). However, these approaches are specifically tailored to this task, whereas the proposed BAE approach should provide a more general method for encoding different constraints.

We specifically evaluate the performance of the proposed BAE approach with respect to capturing distinct cell type-related patterns in different latent dimensions while simultaneously identifying sparse sets of

corresponding marker genes in an application on neurons from the primary visual cortex of adult mice[61], including the analysis of very small groups of cells. We also investigate an adaptation to time-resolved scRNA-seq data, where the aim is to capture developmental patterns in different latent dimensions along with characterizing gene programs, i.e., to infer gene expression dynamics despite no one-to-one correspondence between cells at different time points.

An implementation of our approach, including code to reproduce all results and a tutorial notebook, is available at https://github.com/NiklasBrunn/BoostingAutoencoder.

## Results

### Combining componentwise boosting and neural networks for incorporating structural assumptions into dimensionality reduction

To learn a representation that reflects intuition about underlying structure or experimental design, we have developed the boosting autoencoder (BAE) approach. We adapt an autoencoder-based dimensionality reduction by replacing the encoder by a componentwise boosting approach. This serves to identify a sparse set of genes characterizing each latent dimension and to formalize constraints via the criterion for selecting genes in the boosting. Figure 1 illustrates the overall model architecture, including constraints for disentangled latent dimensions to reflect different cell groups (middle row) and differentiation trajectories via coupling time points (bottom row) as exemplary applications. The normalized gene expression matrix, where rows are cells and columns are genes, is mapped to a latent space via a linear transformation by a sparse weight matrix, obtained from componentwise boosting[43] (see "Componentwise likelihood-based boosting"). The encoder can thus be conceptualized as a linear neural network (i.e., without activation function and bias term) or, alternatively, as a set of generalized linear models (one for each latent dimension) with an identity link function, fitted via componentwise boosting. The decoder neural network maps the latent representation to a reconstruction of the input data (see "Deep learning and autoencoders"). Its parameters are optimized via stochastic gradient descent[62] on the reconstruction loss. After training, a latent representation for downstream analysis is obtained by multiplying the data with the sparse encoder weight matrix.

Boosting as a supervised approach requires a target for model fitting, which corresponds to a criterion for guiding the variable selection. To obtain such a criterion, we use the negative gradients of the autoencoder reconstruction loss with respect to the latent representations, i.e., we couple variable selection and optimization of the autoencoder (see "Core optimization algorithm of the BAE"). The negative gradients correspond to the direction in which the latent representation needs to be changed in order to minimize the reconstruction loss. Thus, variables are selected that are most strongly associated with these directions, ensuring that variable selection aligns with finding an optimal dimensionality reduction. Importantly, the variable selection enforces a sparse mapping to the latent space where only a small number of explanatory variables, i.e., genes, are linked to each latent dimension, facilitating a biological interpretation.

The criterion for variable selection can be flexibly customized by incorporating constraints, corresponding to structural assumptions. Specifically, we developed a disentanglement constraint, where the selection in each dimension considers genes that explain complementary information to that already encoded in other dimensions (Fig. 1, middle row). The approach thus captures distinct sets of genes that characterize the expression patterns of cells in different dimensions (see "The BAE identifies cell types and corresponding marker genes in cortical neurons"), which may, e.g., represent different subgroups of cells, as in "Subgroup analysis of cortical neurons with a small number of cells". As another example for the flexibility of the approach, we developed a constraint for encoding experimental design knowledge on gradually evolving gene expression dynamics in time series data (Fig. 1, bottom row and "Adaptation for time series"). Specifically, we train a BAE with the disentanglement constraint at each time point and pass the weight matrix between time points, such that optimization for a

given time point respects the changes from the previous time point, and dimensions that reflect the same development pattern across time points are coupled. This is to identify developmental trajectories and corresponding gene programs (see sections "Incorporating more complex structural knowledge: timeBAE for temporal patterns" and "Identification of developmental patterns in real-time series scRNA-seq data").

During training, the boosting component updates selected weights with either a positive or negative value, allowing the model to use the entire latent space to learn a well-structured data representation. After training, the gene sets driving each latent dimension can be extracted by inspecting the top nonzero coefficients of the encoder weight matrix. We determine the top genes for each dimension by a change-point criterion (see "Identification of

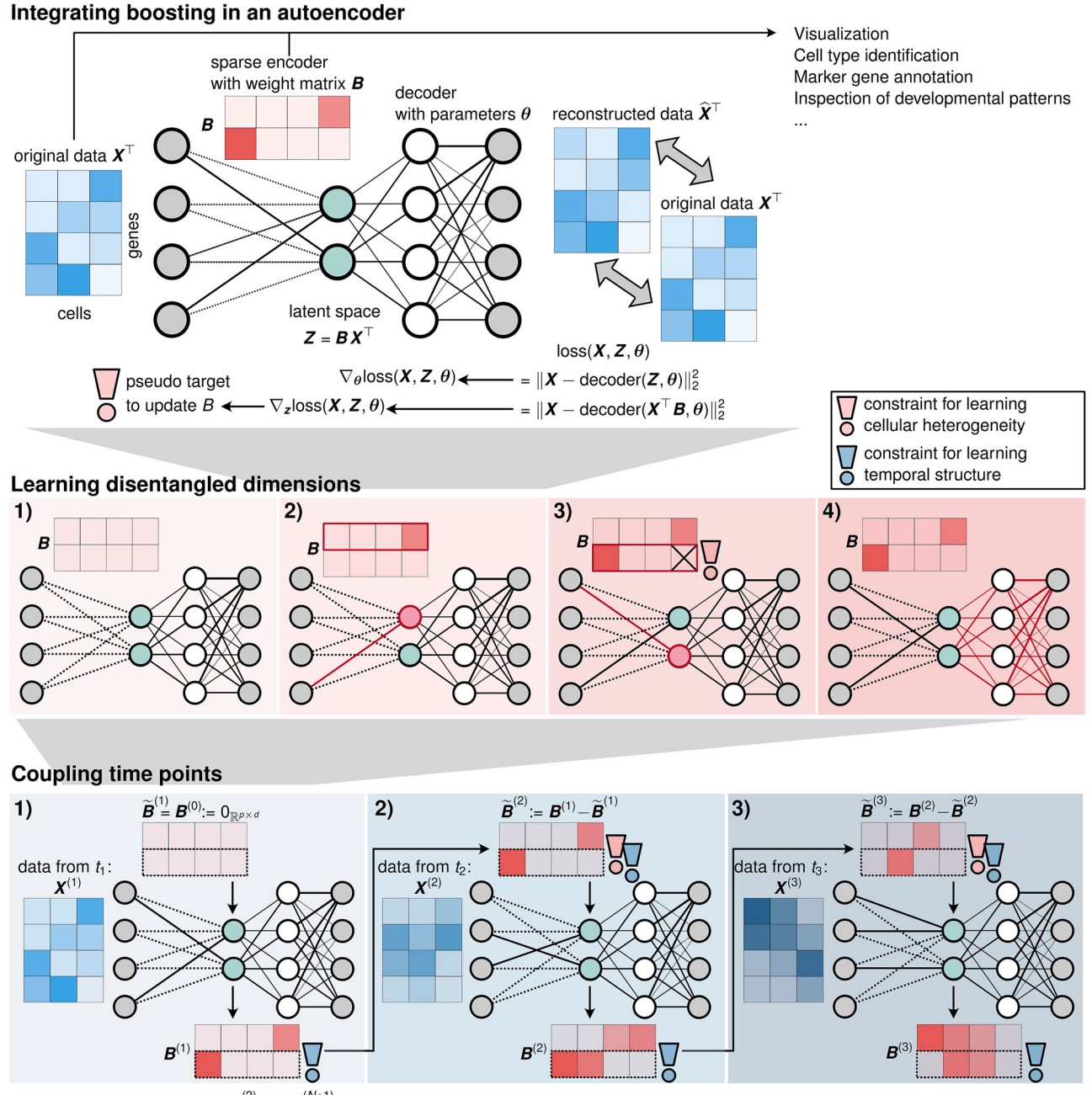

**Fig. 1 | Overview of the boosting autoencoder approach.** Top row: Overview of the boosting autoencoder (BAE) architecture. A gene-by-cell matrix is mapped to a low-dimensional latent space via multiplication by a sparse encoder weight matrix fitted via a componentwise boosting approach, which can be constrained to encode additional assumptions (illustrated by the exclamation mark). Middle row: Training process with a constraint for disentangled latent dimensions. The encoder weights are initialized as zero (1), and for the first latent dimension, one coefficient of $B$ is updated to a nonzero value (2). In all subsequent updates, only variables complementary to the ones already selected may be selected (3, constraint indicated by the exclamation mark). After updating one coefficient for each dimension, the decoder parameters $\theta$ are updated via gradient descent (4). Steps (2)–(4) are repeated in subsequent training epochs. Bottom row: Adding a constraint for coupling time points. For extracting differentiation trajectories from time series scRNA-seq data, a BAE is trained at each time point (1–3). The encoder weight matrix, trained with the disentanglement constraint (red exclamation mark), is passed to the subsequent time point in a pre-training strategy (blue exclamation mark), to couple dimensions corresponding to the same developmental pattern across time (indicated by the dashed box around the second dimension).

top selected genes per latent dimension"). In addition, the representations of cells in particular dimensions can be examined by comparing the signs of the representation values with the signs of the normalized coefficients corresponding to the top selected genes for that dimension. Specifically, if a cell group is highlighted in a latent dimension with positive activation values, the top genes with a corresponding positive weight are characteristic of that group. To simplify the analysis of the learned patterns in distinct latent dimensions, we suggest using a two-dimensional UMAP embedding of the BAE representation of cells, colored individually by the representation values of the cells per dimension, for a visually easy-to-interpret comparison. The analysis of gene sets and corresponding latent patterns can be used to elucidate underlying biological mechanisms, e.g., for de novo cell type identification and annotation or to identify developmental patterns. In addition, the learned latent representation serves as a basis for standard downstream analyses (discussed further in "The BAE identifies cell types and corresponding marker genes in cortical neurons").

## The BAE identifies cell types and corresponding marker genes in cortical neurons

We apply the BAE to a scRNA-seq dataset from ref. 61 of 1679 neurons of mice from the primary visual cortex (see "Data preprocessing for details on the dataset and preprocessing"). The dataset comprises several neuronal subpopulations that have been characterized by the original authors and annotated with corresponding marker genes (ref. 61, Fig. 3, and Supplementary Fig. 12). Our goal is to recover these subpopulations in different dimensions and simultaneously infer a small set of genes for each dimension, which should include marker genes of the corresponding cell type. Accordingly, we tailor the selection criterion in the boosting component of the BAE to select only genes for each dimension are that carry complementary information to the genes already selected in other dimensions (see "Disentanglement constraint for structuring latent representations") to ensure that different dimensions represent subpopulations with distinct molecular signatures.

After preprocessing, the data matrix contained log1p-transformed normalized counts of 1500 highly variable genes in 1525 cells, including neural cell type marker genes and neurotransmitter receptor genes (ref. 61, Supplementary Fig. 15). We randomly sampled 1325 cells as the training dataset, used the remaining 200 cells as test set, and standardized both datasets. We trained a BAE with a 2-layer decoder and 10 latent dimensions (see "Differentiable programming for joint optimization for details"). Note that the cell type and marker gene annotations from[61] were not provided to the model, but were subsequently used as ground truth to evaluate the identified patterns.

Figure 2a shows a heatmap of the latent representation of all cells after training, sorted and colored by annotated cell types together with a 2D UMAP embedding of the BAE representation of the cells. Different dimensions capture specific groups of cells that closely match annotated cell types. In the UMAP plot, cells of the same type cluster together, further confirming that the BAE has captured cell type-characterizing patterns in its latent dimensions.

To highlight learned patterns in specific latent dimensions, we color each cell in the UMAP embedding according to the value of its representation in a BAE latent dimension (Fig. 2b). A high (positive or negative) value for a cell in a particular dimension means that the genes associated with that dimension (with a positive or negative sign) are characteristic of that particular cell. We can thus visualize the patterns captured in each latent dimension (see Fig. 2b for dimensions 1–5 and Supplementary Fig. 1 for dimensions 6–10).

To quantify the correspondence of learned patterns to cell types, we use a quantile thresholding strategy (see "Determining correspondence of latent patterns to cell types"), confirming that each dimension captures a group of cells that mostly corresponds to a specific annotated cell type (see Supplementary Fig. 2). Inspection of the corresponding selected genes, together with their evolution across training epochs, shows that the top genes for each dimension are among the neural marker genes of the respective cell types as

listed in ref. 61, Fig. 3 and Supplementary Fig. 12 (see "Identification of top selected genes per latent dimension", Fig. 2c and Supplementary Note 1, Supplementary Fig. 3 and Supplementary Tables 1 and 2). In addition, we used held-out test data, which we projected onto the learned representation and annotated based on the predominant labels of the closest cell representations in the training data, to verify that this matches the original cell type annotation (see "Label predictions of unseen data", Supplementary Note 2 and Supplementary Fig. 4). We also applied the underlying boosting algorithm (comp$L_2$Boost, see "Componentwise likelihood-based boosting") in a supervised mode using annotated cell type labels as responses. Comparison of the latent BAE patterns with the predictions from the supervised comp$L_2$Boost showed that the BAE identifies very similar patterns despite being unsupervised (see Supplementary Note 3 and Supplementary Fig. 5).

To investigate the robustness of the BAE, we performed a gene selection stability analysis, comparing the total number of genes selected by the BAE, which correspond to the nonzero weights in the encoder weight matrix (see "Gene selection stability analysis"). We trained a BAE for 30 times under identical training conditions, but with different randomized decoder parameter initializations. We found that many of the genes selected in more than 80% of the 30 runs (20/54) were among the neural marker genes and 11 genes were among the nonneural marker genes listed in[61] (Fig. 3 and Supplementary Fig. 12), one of the genes selected in more than 80% of the runs was listed as a neurotransmitter receptor gene in[61] (Supplementary Fig. 15), and the remaining genes were not listed at all. Detailed results are shown in Supplementary Note 4, Supplementary Fig. 6, and Supplementary Table 3. As a follow-up robustness analysis, we utilized information on the selection frequency of each gene in the selection stability analysis to systematically replace the most frequently selected genes with noise genes (see "Robustness analysis of structuring latent representations"). Overall, the BAE demonstrated a robust preservation of the core neural cell structure, even when a substantial fraction of informative genes was replaced with noise (see Supplementary Note 5, Supplementary Fig. 7 and Supplementary Table 4 for detailed results).

The BAE approach is also capable of capturing and characterizing small subgroups of cells. For example, in our application, it identifies two distinct groups of *Sst*-expressing neurons in dimensions 4 and 5, and simultaneously provides candidate marker genes via the sparse gene sets associated with the corresponding dimensions (Fig. 2b, c; more details on the *Sst*-groups in "Subgroup analysis of cortical neurons with a small number of cells"). In addition, we noticed that across several training runs with different decoder parameter initializations and model hyperparameters, one of the latent dimensions consistently captured the same small group of cells of various annotated types, here represented in latent dimension 8 (see Supplementary Fig. 1). The corresponding scatter plot of the top genes suggests *Gpr17* and *Cd9* as characteristic of that group. Both genes are listed as markers of oligodendrocyte precursor cells in the original study of cells from the primary visual cortex of mice (see Fig. 3 and Supplementary Fig. 12 in ref. 61). Furthermore, the receptor Gpr17 has been identified as a a regulator of oligodendrocyte development (see, e.g., refs. 63,64). However, the corresponding cells were classified as different neurons in ref. 61.

While cell identities are often determined in a two-step process, where clustering of cells is followed by marker gene selection, e.g., based on differential expression, the BAE can simultaneously identify groups and a corresponding sparse gene expression signature in a one-step approach. Because it does not rely on predefined cell clusters or types and can capture very small groups of cells, it can identify misclassified cells, help to determine the identity of unclassified cells or investigate cell types at a finer granularity. As shown so far, the BAE allows to encode expected structural assumptions about the presence of different subpopulations of neurons with different sets of transcriptomic markers, and facilitates biological interpretability by allowing to detect cell group-specific gene sets even for small cell groups. Its key benefits are the flexibility to encode structural assumptions and the identification of sparse interpretable characteristic gene sets.

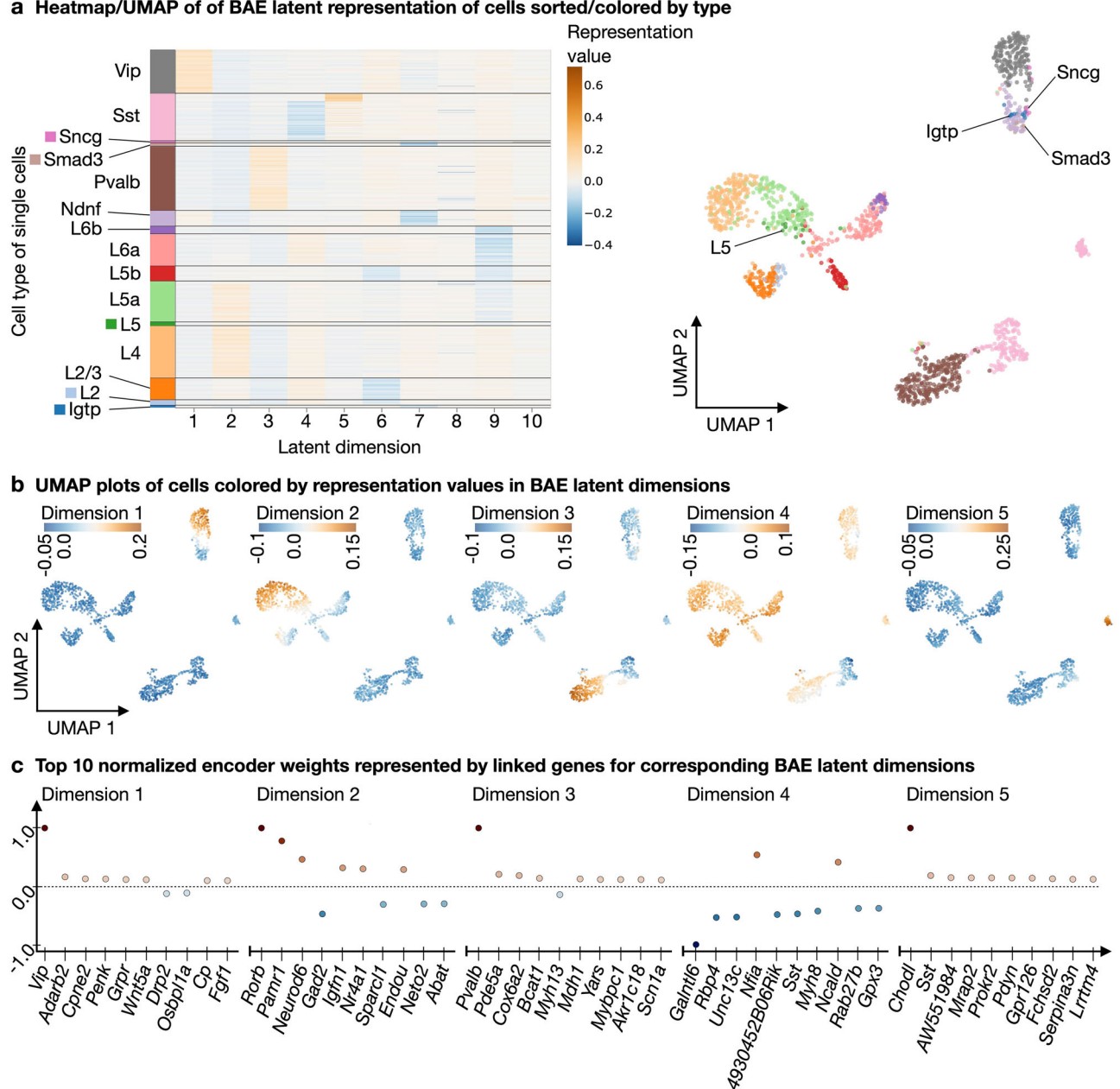

**Fig. 2 | BAE analysis on cortical mouse scRNA-seq data. a** Left: Heatmap of a sparse BAE latent representation of cortical neurons sorted by cell type. Right: Scatter plot of a 2D UMAP embedding of the sparse representation colored by types. **b** Scatter plots of the same UMAP embedding, colored by the representation values of cells in individual BAE latent dimensions. The first five dimensions are shown.

**c** Scatter plots show the top ten normalized nonzero BAE encoder weights (represented by linked genes) in decreasing order of absolute value (from left to right) per latent dimension. The first five dimensions are shown. Normalization was performed by dividing each weight by the maximum absolute weight value for the corresponding latent dimension.

To show that these advantages can be obtained without loss of performance in downstream analyses, we exemplarily compare the BAE representation to a PCA representation in a standard downstream scRNA-seq workflow. Specifically, we use both representations as a basis for visualization and clustering. UMAP embeddings of a BAE representation show a very similar structure to the UMAP based on the PCA representation, grouping together cells of the same type (Supplementary Fig. 8). In addition, we performed Leiden clustering based on the BAE and PCA representations and computed silhouette scores and adjusted Rand indices for 20 clustering results for each approach, where the BAE was re-trained with a different random initialization of the decoder parameters for each clustering (see "Clustering analysis of BAE and PCA based representations of cortical neurons for details"). The median silhouette score for the BAE

clusterings is 0.41 and the median silhouette score for the PCA clusterings is 0.43, i.e., there is no substantial difference in performance when taking into account the variability of the BAE due to random initializations (interquartile range of 0.03). The median adjusted Rand index between the two clusterings is 0.82. These results support the use of the BAE as an alternative dimensionality reduction, as it provides competitive performance to the standard approach while offering the additional benefits of a sparse gene set selection and the flexibility to structure the latent representation based on task-specific criteria.

We also compared the BAE to two other unsupervised variable selection methods, sparsePCA[40] and scPNMF[65], in terms of gene selection (see "Model comparison for gene selection for details"). The BAE encoder weight matrix exhibited the highest sparsity level among all approaches,

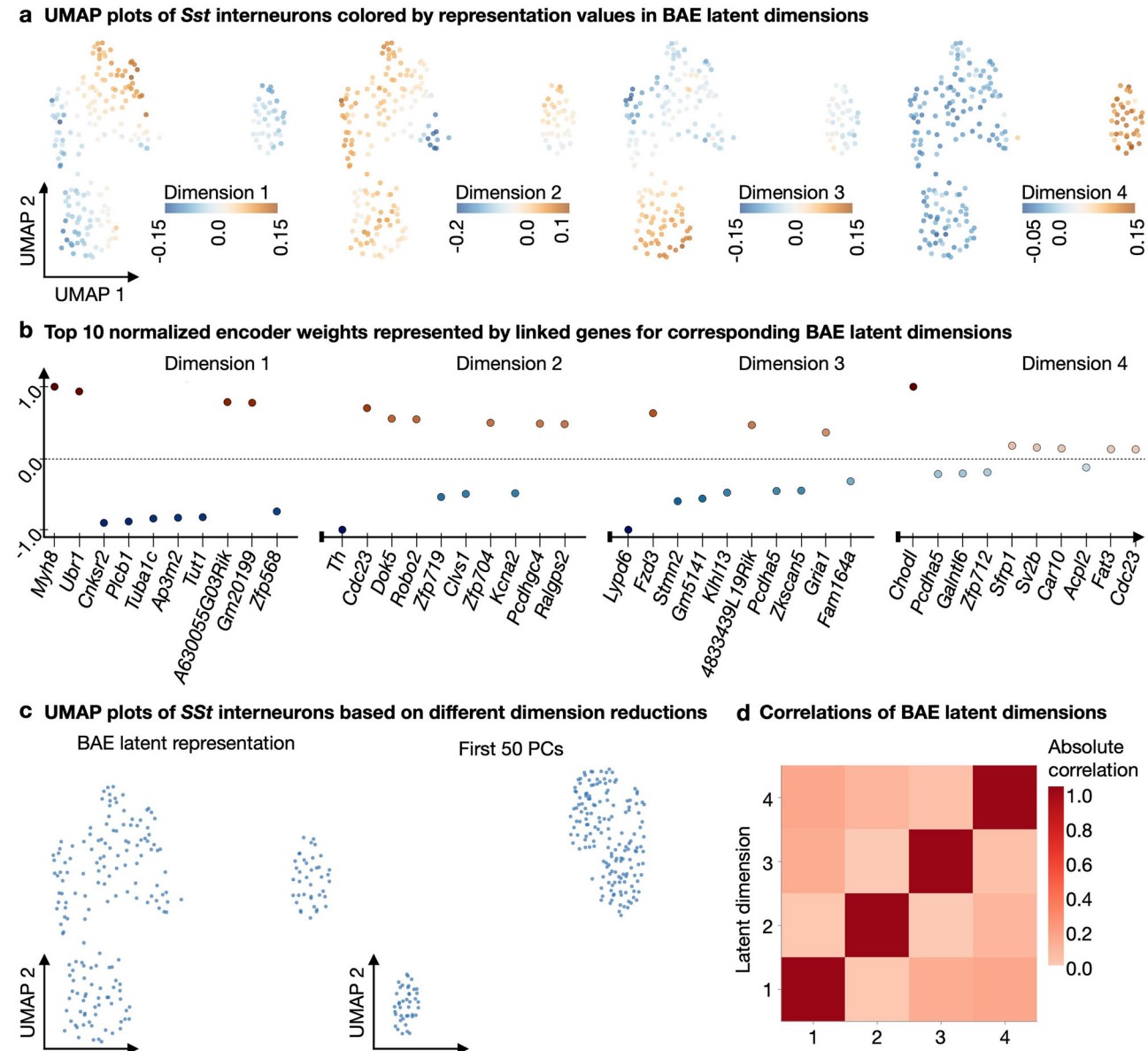

**Fig. 3 | BAE subgroup analysis of *Sst*-expressing neurons from mouse cortex.**
**a** Scatter plots of a 2D UMAP embedding based on a 4D BAE latent representation of *Sst*-neurons, colored by the representation values of the cells in each dimension. **b** Scatter plots showing the top ten normalized nonzero BAE encoder weights (represented by linked genes) in decreasing order of absolute value (from left to right) per latent dimension. Normalization was performed by dividing each weight by the maximum absolute weight value for the corresponding latent dimension. **c** Left: Scatter plot of the 2D UMAP embedding calculated using the 4D BAE latent representation. Right: Scatter plot of a 2D UMAP embedding calculated using the first 50 principal components resulting from PCA of the *Sst*-neurons. **d** Heatmap shows the absolute values of the Pearson correlation coefficients between the representations of cells in the BAE latent dimensions.

reflecting the fewest gene-to-latent-component connections. The analysis results are detailed in Supplementary Note 6 and Supplementary Table 5.

## Subgroup analysis of cortical neurons with a small number of cells

Motivated by the fact that the BAE was able to detect a subset of *Sst*-expressing neurons characterized by the neural marker *Chodl*, we singled out the *Sst*-expressing neurons for a more fine-grained detection of *Sst*-expressing subgroups. This is to demonstrate that the BAE can be useful even with a very small number of cells. Here, subgroup detection is challenging because there are only very few *Sst*-neurons, namely $n = 202$. Also, cells of the same type tend to be more homogeneous in their expression profile and, thus, may be more difficult to disentangle. To identify subgroups, we select the 1500 most highly variable genes on the subpopulation

and train a BAE using the disentanglement constraint with four latent dimensions for 20 epochs (see "Statistics and reproducibility and Data preprocessing").

After training, we again inspect the learned patterns based on UMAP plots of the latent embedding, colored by the representation values in different latent dimensions, together with scatter plots of the corresponding selected top genes for the dimensions (see Fig. 3a, b). For comparison, we computed a UMAP embedding based on a PCA representation of only the *Sst*-expressing neurons, which visually does not separate distinct clusters as well (see Fig. 3c). Examining the patterns learned in different latent dimensions of the BAE via the colored UMAP plots shows that the approach captures distinct small subgroups of cells. Inspection of the sample Pearson correlation coefficients between the latent dimensions confirms that the learned dimensions of the BAE are uncorrelated, thus different cell group-

characterizing genes were associated with different dimensions (see Fig. 3d). By analyzing the selected genes in the scatter plots, we observed that several genes with the highest absolute weight values per latent dimension correspond to those listed in ref. 61, Fig. 3 and Supplementary Fig. 12, for subgroups of *Sst*-neurons. For example, the subgroup of *Sst*-neurons characterized by the gene *Chodl* is captured in dimension 4 and clearly separated from the remaining cells. Also, for dimension 1, *Myh8* was the top selected gene characterizing a subgroup at the top center of the UMAP plot. In dimension 2, the top selected gene is *Th*, which characterizes a small group of cells in the center of the UMAP plot.

### Incorporating more complex structural knowledge: timeBAE for temporal patterns

To demonstrate how the BAE approach can be flexibly tailored to different data structures via the variable selection criterion, we investigate an application on simulated time series scRNA-seq data. The goal is to (1) identify distinct developmental patterns at each time point, driven by corresponding sets of genes, and (2) link latent dimensions that capture the same developmental pattern at different time points. This will then allow the identification of distinct trajectories across time points, despite the lack of one-to-one correspondence between cells at different time points. We assume that multiple differentiation trajectories may be present in the data. Specifically, we expect the presence of different groups of cells at each time point, where each group at one time point may be linked to another group at the following time point. Linked groups are characterized by some similar, but also some novel gene expression patterns, i.e., there is some overlap in the characterizing gene sets at subsequent time points, corresponding to the gradual up- and downregulation of genes during development. We illustrate how these assumptions can be incorporated into the BAE approach, and show how this adaptation can be evaluated by a corresponding simulation.

We simulate data to reflect the developmental processes of three groups of cells across three time points. Each group is characterized by a distinct, small set of genes at each time point, with some overlap of genes from linked groups at two subsequent time points, as shown in Fig. 4a. Specifically, we generate binary counts indicating high vs. low expression of a gene, similar to the simulation design in ref. 35. Each group is characterized by 8 highly expressed genes, 3 of which are also highly expressed in the subsequent group of cells along the same differentiation trajectory at the next time point (e.g., genes 1–8 are characteristic for group 1 at time point 1, genes 6–13 for group 1 at time point 2, genes 11–18 for group 1 at time point 3, etc.). In addition to the in total 54 trajectory-characterizing genes, we simulate 26 non-informative noise genes. The simulated expression value for each cell is generated by sampling from a Bernoulli distribution with a higher probability of success for a highly expressed gene, and a lower probability of success for a lowly expressed gene (see "Simulation of scRNA-seq-like data with time structure for details"). For our analysis, we generated count matrices consisting of 310 cells at the first time point, 298 cells at the second time point, and 306 cells at the third time point, equally dividing the cells among the three groups.

To not only infer the characterizing genes for each group and time point separately but also to identify the groups of cells that overlap in their gene sets, we adapt the BAE to couple dimensions across time points. This allows the identification of cells that belong to the same differentiation trajectory, knowing that an overlap in gene sets does not exclusively correspond to cell differentiation. Specifically, we train one BAE for each time point, using the same number of dimensions for each model (see "Adaptation for time series for details"). For example, assuming three distinct groups at three time points, we train three BAEs with three latent dimensions each.

The overall approach, which we call timeBAE, is optimized by successively training each BAE with the data from one time point. While the decoders are randomly initialized, the encoder weights for a given time point are initialized with the learned encoder weights from the previous time point. That is, when we optimize the encoder at one time point, we simultaneously pre-train the encoder of the following time point, as shown

in Fig. 1. Thus, we guide the BAE of each time point to preferentially select variables related those already included in the same latent dimension at a previous time point, in order to capture gradually evolving gene patterns that characterize distinct developmental trajectories. For the final latent representation, we concatenate the representations of the individual BAEs. The overall approach reflects cell group-characterizing patterns at each time point, while tracking the developmental processes of these groups via the correspondence of latent dimensions across time. The approach is compatible with both the alternating and the joint training mode explained in "Core optimization algorithm of the BAE".

The results for training a timeBAE with three latent dimensions for each BAE at each time point on the simulated data are shown in Fig. 4. Panel a shows heatmaps of the final BAE encoder weight matrices after training on the data at every single time point. We designed the timeBAE such that the same latent dimension in each individual BAE represents the same cell lineage over time. Sparse sets of selected genes for each dimension correspond to the simulated developmental patterns of highly expressed genes over time, shown by the vertical dashed lines in Fig. 4a. A direct comparison of the whole binary count data with its latent representation by the timeBAE encoder further shows that the sparse patterns of the representation match the patterns in the simulated binary count data (Fig. 4b). In particular, cells in a latent dimension that encodes patterns of a specific time point in a lineage are slightly activated when the cells correspond to measurements from the previous or subsequent time point. In addition, the correlation patterns of the latent representation in Fig. 4c show that almost all latent dimensions representing the same developmental process share a higher absolute Pearson correlation coefficient compared to other dimensions.

To highlight the importance of pre-training encoders for linking latent dimensions over time, we compared this approach to a simpler strategy where encoder parameters are initialized to zero at the start of each BAE training. It can be seen that the model with the zero initialization strategy is no longer able to link corresponding latent dimensions across time (Supplementary Fig. 9).

### Identification of developmental patterns in real-time series scRNA-seq data

In a further experiment, we apply the timeBAE approach to a real-time series scRNA-seq dataset of human embryonic stem cells grown as embryoid bodies over a period of 27 days, where differentiation into different cell lineages can be observed[66]. Cells were sequenced during 5 consecutive time intervals, labeled by time points 1–5.

We aim to capture sparse gene sets associated with different latent dimensions that may reflect potential markers of different cell stages in a cell lineage. To validate our approach, we consider a subset of the data where the learned patterns in the linked dimensions can be easily visualized and compared to an embedding that reflects both cluster and temporal structure. Finding such an embedding, where both temporal structure and cluster identity is simultaneously visualized in 2D, is challenging[66]. Therefore, we trim the dataset to retain only cells that can be grouped into large clusters, with a temporal gradient of cells from different time points within each cluster (see "Data preprocessing"). After preprocessing, we ended up with 11,627 cells from 3 different time points with 3 different annotated cluster labels and 262 pre-selected genes for subsequent analysis. Figure 5a shows a 2D UMAP embedding of the trimmed and clustered dataset, which allows to visualize both cluster and time structure.

Next, we train a timeBAE with a 2-layer decoder for each BAE at the different time points using the disentanglement constraint. Each BAE is defined to have four latent dimensions. Thus, the timeBAE has a total of 12 dimensions. After training, we rearrange the dimensions such that linked dimensions corresponding to different time points are adjacent to each other to facilitate the interpretation of the visualized results. The reorganization is described in "Adaptation for time series".

In Fig. 5b, we show exemplary UMAP plots colored by timeBAE latent dimensions 1–3 after reordering, where dimension 1 corresponds to the learned patterns from the first time point, dimension 2 corresponds to the

## a  Simulated time series scRNA-seq data with timeBAE encoder weight matrices at different time points

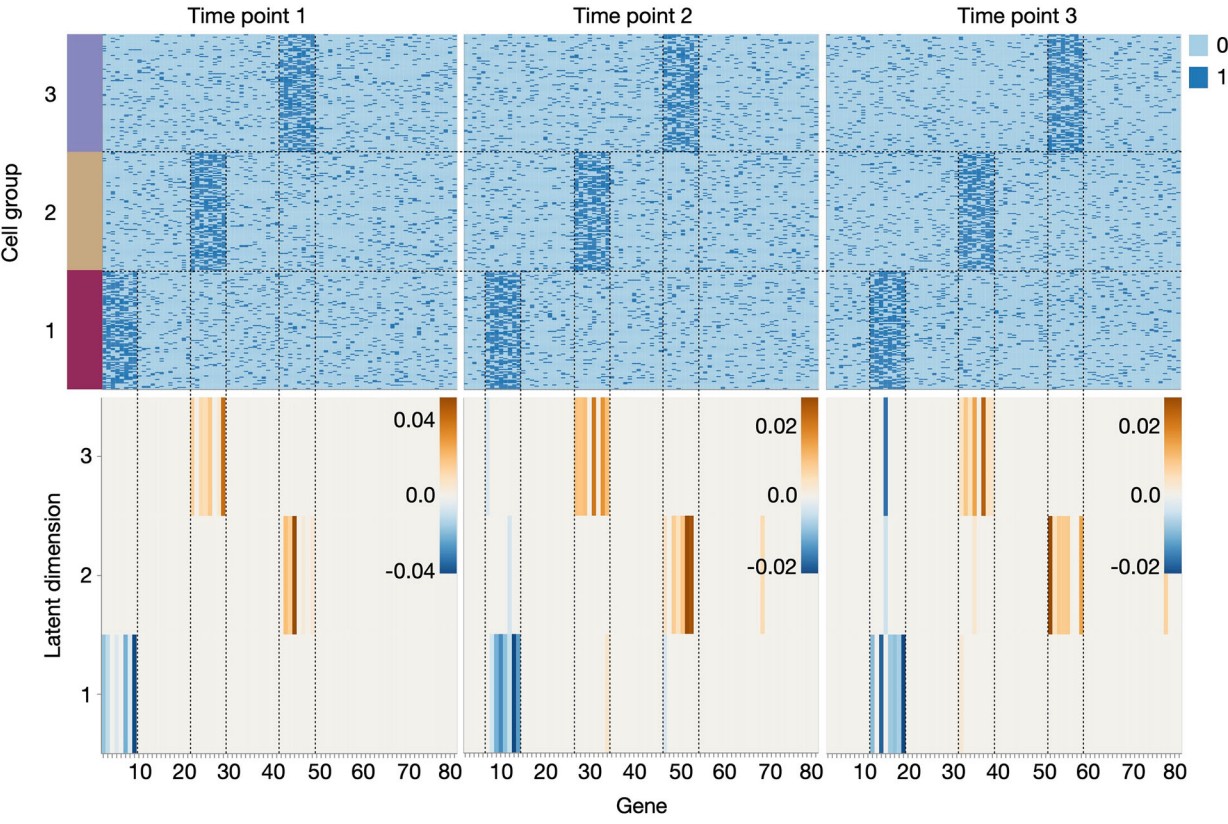

## b  Data at all time points with timeBAE latent representation

## c  Correlations of latent dimensions

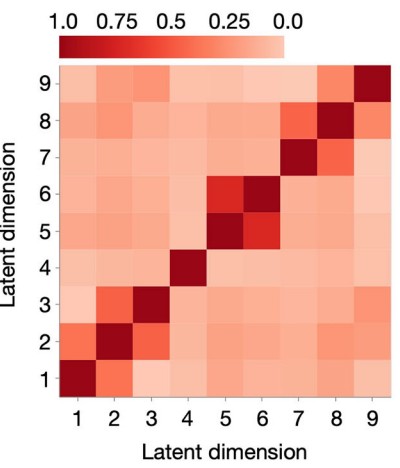

**Fig. 4 | Investigation of the timeBAE approach on simulated dichotomized scRNA-seq data. a** Top row: Heatmaps show the simulated binary count matrices mimicking three lineages of cells across three different time points. Bottom row: Heatmaps illustrate the learned timeBAE encoder weight matrices per time point. **b** Heatmaps of the the stacked binary count data of all time points and the timeBAE latent representation. **c** Heatmap of the absolute values of sample Pearson correlation coefficients between the timeBAE latent dimensions.

second, and dimension 3 corresponds to the third. Comparing the learned patterns with the original cluster and time of measurement information shows that the timeBAE approach captured genes that characterize temporal subgroups of cluster 3 in the correct order. The corresponding set of genes, sorted by importance, is shown in Fig. 5c and could provide marker gene information for different cell stages in a cell lineage. However, the learned patterns did not match the cluster and time point patterns from Fig. 5a for every set of linked dimensions (Supplementary Figs. 10 and 11). In general, the subsequent analysis of temporal patterns in cells potentially revealed by the timeBAE approach should be carefully performed based on

the top selected genes for the linked dimensions. This typically requires prior biological knowledge of known markers of cell lineages.

In such a setting, where both temporal structure and cluster identity can be identified in a 2D visualization, we can compare the timeBAE results to a linear regression analysis where we use the time point and cluster annotations to determine the most significant genes and compare them to the top genes selected by the timeBAE (see "Gene selection comparison for the timeBAE"). We determined significant genes in the regression analysis by coefficients with the smallest adjusted p-values after correction for multiple testing. Each set of significant genes for each cluster at each time

### a  UMAP plots of embryoid body cells

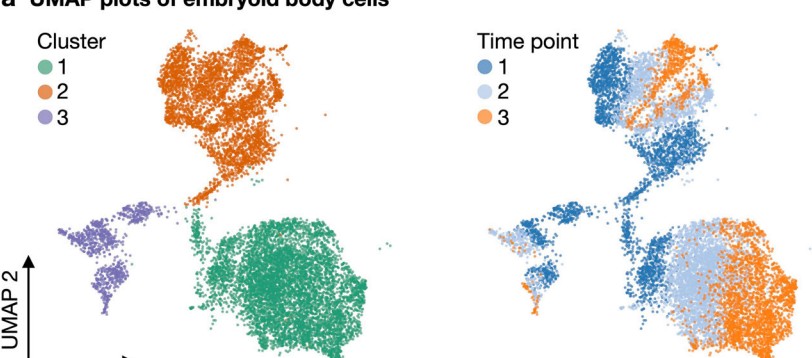

### b  UMAP plots of embryoid body cells colored by representation values in linked timeBAE latent dimensions

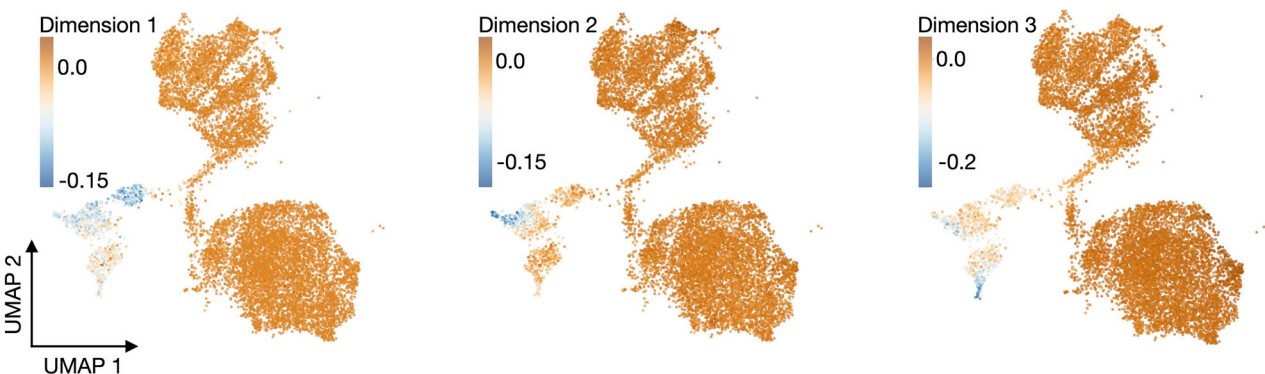

### c  Top 10 normalized encoder weights represented by linked genes for corresponding timeBAE latent dimensions

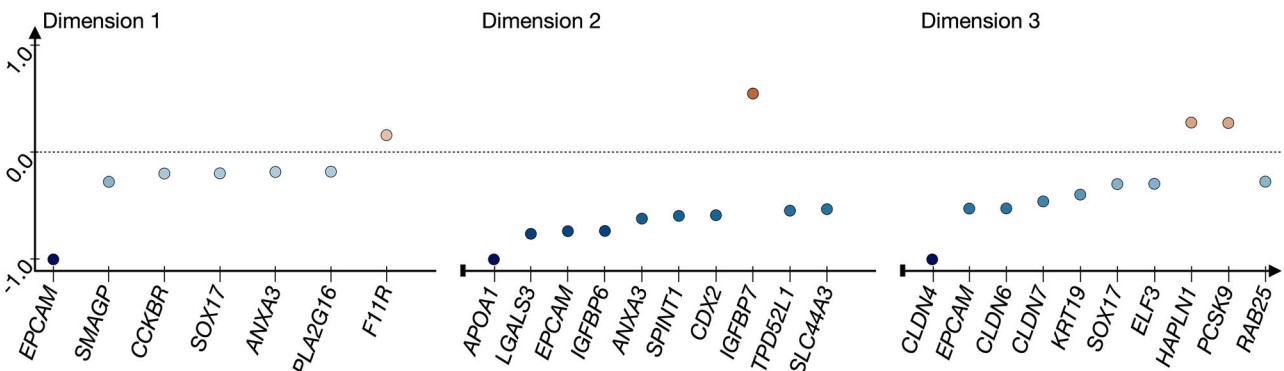

**Fig. 5 | Application of the timeBAE approach to embryoid body data. a** Left: Scatter plot illustrates a 2D UMAP embedding of the PCA representation of the trimmed embryoid body data. Individual cells are colored by their cluster label resulting from Leiden clustering with a resolution parameter of 0.025. Right: Scatter plot shows the same embedding colored by measurement time points, each corresponding to a time interval. **b** Scatter plots of the same UMAP embedding, colored by the representation values of cells in linked timeBAE latent dimensions across time. The first three dimensions are shown. **c** Scatter plots show the top ten normalized nonzero timeBAE encoder weights (represented by linked genes) in decreasing order of absolute value (from left to right) per latent dimension. The first three dimensions are shown. Normalization was performed by dividing each weight by the maximum absolute weight value for the corresponding latent dimension. The first dimension has only seven corresponding nonzero weights.

point had multiple overlaps with a set of top selected genes corresponding to a latent dimension of the timeBAE, confirming the importance of the top selected genes for time point-related patterns per cluster (Supplementary Note 7 and Supplementary Table 6). Note that we deliberately created this setting, where clusters of cells, which each corresponds to a distinct developmental trajectory, can be visually identified in a standard 2D UMAP plot. This enables us to apply the regression as a simpler, supervised approach, serving as a surrogate of a ground-truth annotation of lineage trajectories. The regression analysis is thus intended not as a performance competitor, but as a means to provide a reference annotation to verify that the timeBAE results are meaningful, which is only feasible in the simplified setting. In

addition, the results suggest that the timeBAE approach can also be applied in a scenario where cluster identity and temporal variation cannot be as easily disentangled in a 2D space. While a supervised approach would no longer be applicable in such a more realistic setting, the timeBAE would still be able to link characterizing gene sets to related latent dimensions that could be used for further post-hoc analysis of lineage relationships.

## Discussion

Low-dimensional representations of single-cell data allow for visualization of the underlying cellular heterogeneity. We have developed the BAE approach to additionally incorporate structural assumptions, e.g., based on

expected patterns or experimental design, directly into the dimensionality reduction. The approach combines variable selection via componentwise boosting, where the selection criterion can encode constraints reflecting assumptions, with neural network-based autoencoders for dimensionality reduction. Each latent dimension is associated with a sparse set of genes that characterizes the pattern captured by the variable selection component in that dimension, allowing for a biological interpretation.

To illustrate the flexibility of such an approach for targeted dimensionality reduction in different scenarios of scRNA-seq analysis, we have developed and illustrated two types of constraints. A constraint for disentangled dimensions reflects the intuitive assumption that different groups of cells (e.g., belonging to the same type or cell stage) are characterized by non-overlapping gene sets, while a constraint capturing different developmental trajectories in time series scRNA-seq data reflects experimental design knowledge about gradual changes in gene expression patterns between time points. Accordingly, we have applied the BAE approach to capture known marker genes of neurons, reflecting the assumption that distinct groups are present in the data, and to recover differentiation trajectories. The BAE partitioned distinct cell types in uncorrelated latent dimensions along with a sparse set of biologically meaningful marker genes in an application on the subset of neurons from the primary visual cortex of mice, and allowed the recovery of distinct differentiation lineages in simulated and real-time series data. While we developed the second constraint to analyze time series scRNA-seq data, the approach could be more generally applicable to study changes in gene sets of cell groups between conditions, e.g., health and disease.

Notably, the approach simultaneously infers cell groups and characterizing genes, in contrast to standard two-step approaches, where clustering and differential gene expression analysis for marker gene annotation are performed separately. Our joint approach allows even small groups of cells to be identified and characterized by their associated gene sets. In an exemplary application, we were able to resolve the identity of a small group that was not captured by the original annotation, and recover several subtypes of *Sst*-expressing neurons in a subgroup analysis, illustrating the potential of incorporating constraints into scRNA-seq dimensionality reduction.

We have presented these different applications to showcase the adaptability of the BAE approach to different scenarios, which we see as a key advantage of the method. In addition, we have highlighted in particular its capability to infer biologically meaningful sparse sets of genes to explain each latent dimension, which we have investigated in detail, e.g., by providing a change-point strategy to select the most informative genes and different robustness analyses.

Of course, for each specific type of constraint within the BAE approach, such as the sparsity or disentanglement constraint, there may be several alternative approaches. Rather than benchmarking of each constraint variant individually, our focus was on investigating the usefulness of the overall framework for facilitating a biological understanding in different application scenarios. Nevertheless, we provide an exemplary comparison with alternative approaches on simulated data (Supplementary Note 8–10 and Supplementary Figs. 12–19) and real data (Supplementary Note 6 and Supplementary Table 5), including illustrations of how the encoder weights, corresponding to selected genes, evolve during training. Using the simulation design from ref. 35, we show that the BAE is the only autoencoder-based approach that simultaneously provides a sparse and disentangled representation without additional loss function penalty terms (Supplementary Note 10 and Supplementary Figs. 14–19). We also compared the BAE representation to PCA as a basis for standard downstream analyses, not for establishing superiority, but rather to verify that the unique advantages of the BAE can be obtained without compromising performance.

In the time series example, we subset the dataset so that both cluster and temporal structure could be captured simultaneously in a 2D UMAP visualization. While this simplification might seem somewhat artificial, it allowed us to use a simple regression approach to provide a reference annotation, to thus evaluate whether the timeBAE can correctly identify meaningful structure. Naturally, in such a simplified scenario, a simpler approach might also be able to pick up genes within each cluster that are differentially expressed between time points. Yet, our approach is more generally applicable also in more complex settings where the underlying patterns cannot be visualized in 2D. Then, it may represent multiple lineages in the same set of coupled latent dimensions, which requires more detailed post-hoc analyses of the identified trajectories.

While the BAE approach is relatively robust to the number of latent dimensions, some intuition about the expected number of different groups in the data is required to set the number of dimensions within a suitable range. In addition, when fitting time series data, we currently assume a constant number of groups over time. To also allow for branching trajectories, which are typical for many differentiation processes, one dimension at an earlier time point could be linked to several dimensions at a later time point, e.g., by tailoring the penalization in the boosting coefficient updates. In addition, the BAE in its current form has a linear encoder, which can limit flexibility and affect reconstruction performance. This might cause problems in particular for complex and very large datasets, where a greater model capacity would be needed. While we found the simple single-layer encoder sufficient in the applications, an extension to include a nonlinear activation function and additional layers to allow for higher complexity is in principle feasible. A nonlinear activation function may require to extend the boosting to a generalized linear model framework, as discussed in ref. 43, and choosing an appropriate link function. To include additional layers, our joint optimization strategy based on differentiable programming would allow differentiation through the boosting layer to obtain gradient feedback for the previous layers. However, the boosting coefficients would then not directly correspond to the contribution of individual input variables, but to a (potentially nonlinear) combination of them. Variable attribution methods, such as those discussed in refs. 67,68, could be used to solve this problem.

Regarding scalability, the BAE in principle is also applicable to larger datasets. However, a key focus in the present manuscript is the applicability of the BAE to small data scenarios, e.g., for identifying and characterizing rare cell types. We have thus not specifically investigated scalability and have not focused on this in the current implementation. Yet, scaling up the approach to large-scale datasets would be an exciting future extension to make the approach even more widely usable.

As the BAE is already inherently regularized by the boosting component, which enforces sparsity, we have not experimented with further regularizations so far. However, investigating regularization such as dropout or decoder weight decay would be an interesting direction for future research, in particular when extending the approach to more complex architectures.

In our application, we have focused on highly variable genes as model inputs. However, a key advantage of boosting is that it can handle very large numbers of variables[45]. The approach could thus be scaled to even higher-dimensional input data, where approaches from ref. 69 could be employed for computational efficiency. This could also be useful, for example, for single-cell ATAC-sequencing data on chromatin accessibility, which is more sparse and high-dimensional than scRNA-seq data. The approach could then also be extended to a multi-omics setting, where a latent representation is learned that integrates different modalities, and the selection criterion could be adapted to capture modality-specific or modality-overlapping information in different dimensions, e.g., by considering residuals of model fits from one modality for fitting the other modality. Further, additional knowledge about the correspondence between peaks and genes could be incorporated by adjusting the penalty for the corresponding boosting coefficient updates, to increase the probability of selecting these variables together[45,70]. Similar strategies could be used to incorporate knowledge about pathway structure or gene regulatory networks.

Another exciting possible strategy for incorporating additional information could be the extension of the BAE to a graph-based framework. In particular, recently graph neural networks and graph attention networks have gained popularity for embedding neighborhood information when

modeling gene expression data, by incorporating local information about cells with similar expression profiles or spatial niche information from spatial transcriptomics[71–73]. Such approaches could be incorporated into the BAE framework, e.g., by adapting the encoder to a single-layer graph attention or graph convolution layer. The boosting component could then be integrated to learn a sparse weight matrix for mapping cells to their value representation, and local neighborhood information about spatially close or transcriptionally similar cells could be leveraged to further strengthen the sparse patterns learned in the latent representation.

## Methods

### Mathematical notations

Matrices are written in capital bold letters, and vectors in lowercase bold letters. Greek letters are used for denoting model parameters or hyperparameters. Vectors are treated as column vectors. Scalar values are denoted by lowercase letters. In general, $X \in \mathbb{R}^{n \times p}$ denotes a matrix of $n$ i.i.d. observations (cells) of $p$ variables (genes) sampled from an unknown probability distribution. Observations, i.e., rows of $X$, are denoted by $x_1, \ldots, x_n$ and the columns of $X$ by $x^{(1)}, \ldots, x^{(p)}$. We define the zero-matrix with $n$ rows and $p$ columns by $\mathbf{0}_{\mathbb{R}^{n \times p}}$. Further, by $\widehat{\mu}_x := \frac{1}{n} \sum_{i=1}^{n} x_i$ we denote the sample mean of a vector $x = (x_1, \ldots, x_n)^\top$ and by $\widehat{\sigma}_x := \sqrt{\frac{1}{n-1} \sum_{i=1}^{n} (x_i - \widehat{\mu}_x)^2}$ the corrected sample standard deviation of $x$. The sample Pearson correlation coefficient of two vectors $x, y$ is denoted by $\widehat{\mathrm{cor}}(x, y) := \frac{\sum_{i=1}^{n} (x_i - \widehat{\mu}_x) \cdot (y_i - \widehat{\mu}_y)}{\widehat{\sigma}_x \widehat{\sigma}_y}$. Given a vector $x$ we refer to its standardized version by computing the z-score for each component $\widehat{x} = (\widehat{x}_1, \ldots, \widehat{x}_n)^\top$, where $\widehat{x}_i := \frac{x_i - \widehat{\mu}_x}{\widehat{\sigma}_x}$, $i = 1, \ldots, n$. Furthermore, given a matrix $X \in \mathbb{R}^{n \times p}$ and a set $\mathcal{I} \subset \{1, \ldots, p\}$ with $|\mathcal{I}| < p$, we denote by $X^{(-\mathcal{I})}$ the sub-matrix consisting of only the columns with indices that are not elements of $\mathcal{I}$. In the special case of $\mathcal{I} = \emptyset$, $X^{(-\mathcal{I})} = X$.

### Deep learning and autoencoders

Deep learning models are based on artificial neural networks (NNs)[74]. Technically, NNs correspond to function compositions of affine functions and so-called activation functions, organized in layers. Formally, for $m, n_0, \ldots, n_m \in \mathbb{N}$ with $m \geq 2$, given some differentiable and typically nonlinear functions $\rho^{(i)} : \mathbb{R} \mapsto \mathbb{R}$, weight matrices $W^{(i)} \in \mathbb{R}^{n_i \times n_{i-1}}$ and bias vectors $b^{(i)} \in \mathbb{R}^{n_i}$, $i = 1, \ldots, m$, a NN is a function composition

$$f : \mathbb{R}^{n_0} \mapsto \mathbb{R}^{n_m}$$
$$f(x) := (f^{(m)} \circ \cdots \circ f^{(1)})(x),$$

where for $i = 1, \ldots, m$ the functions $f^{(1)}, \ldots, f^{(m)}$ represent the individual layers of the NN, defined by

$$f^{(i)} : \mathbb{R}^{n_{i-1}} \mapsto \mathbb{R}^{n_i}$$
$$f^{(i)}(x) := \rho^{(i)}_{(\cdot)}(W^{(i)} x + b^{(i)}).$$

Thereby, ( · ) in the subscript of an activation denotes its elementwise application to the input vector.

The deep architecture of NNs allow for modeling complex relationships (e.g., see refs. 75–77). The search for optimal weights and biases, i.e., the training of the NN, is usually performed via gradient-based optimization schemes. Therefore, given a fixed observation vector $x \in \mathbb{R}^{n_0}$, a NN can be equivalently considered as a mapping from a finite-dimensional parameter space $\Theta := \mathbb{R}^{n_0 \times n_1} \times \mathbb{R}^{n_1} \times \ldots \times \mathbb{R}^{n_{m-1} \times n_m} \times \mathbb{R}^{n_m}$ to the output space $\mathbb{R}^{n_m}$

$$NN : \Theta \mapsto \mathbb{R}^{n_m},$$
$$NN(x; \theta) := f(x; \theta) = (f^{(m)}_{W^{(m)}, b^{(m)}} \circ \cdots \circ f^{(1)}_{W^{(1)}, b^{(1)}})(x),$$

where $\theta = (W^{(1)}, b^{(1)}, \ldots, W^{(m)}, b^{(m)})$ represents a tuple of weight matrices and bias vectors.

An autoencoder(AE) is a specific neural network architecture with a bottleneck-like structure frequently used for unsupervised representation learning of high-dimensional data. Specifically, AEs consist of an encoder network, which defines a parametric mapping from the observation space into a lower-dimensional latent space, and a decoder network, which performs the reverse mapping back to data space, each parameterized by a tuple of tunable weight matrices and bias vectors

$$AE : \Theta \mapsto \mathbb{R}^{n_0},$$
$$AE(x; (\psi, \theta)) := f_{\mathrm{dec}}(f_{\mathrm{enc}}(x; \psi); \theta).$$

To learn a low-dimensional representation of the observed data that reflects the main underlying characteristics, the training objective for AEs is to approximate the identity function under the constraint that the function input first gets compressed to a low-dimensional space before the reconstruction. This requires a differentiable scalar valued reconstruction loss function $L_{\mathrm{rec}} : \mathbb{R}^{n_0} \mapsto \mathbb{R}^{n_0}$ which measures the similarity of the reconstructed output to the original input. A common choice for continuous observation data is the mean squared error

$$L_{\mathrm{rec}}(x, \widehat{x}) := \frac{1}{p} \sum_{j=1}^{p} (x_j - \widehat{x}_j)^2.$$

For normally distributed data, minimizing the reconstruction loss, i.e., the mean squared error, then corresponds to maximum likelihood estimation of the network parameters.

Given observations $x_1, \ldots, x_n \in \mathbb{R}^{n_0}$, the search for optimal weight matrices and bias vectors of the AE is performed by iteratively minimizing the empirical risk

$$\underset{(\psi, \theta) \in \Theta}{\arg \min} \frac{1}{n} \sum_{i=1}^{n} L_{\mathrm{rec}}(x_i, AE(x_i; (\psi, \theta))), \tag{1}$$

via a gradient-based optimization algorithm such as stochastic gradient descent. Starting with a random initialization of the weight matrices and bias vectors, in each iteration the current parameters are adapted into the direction of the negative gradient of a randomly determined partial sum of the objective in (1) w.r.t. the weight matrices and bias vectors. To avoid making too large steps into the direction of the negative gradients and to stabilize the optimization, the gradients are multiplied by a scalar factor $v \in (0, 1)$. An advanced version of stochastic gradient descent is the Adam optimizer[62], a computationally efficient first-order gradient-based method with an adaptive learning rate incorporating additional first- and second-order moments of the gradients.

After training an AE, the observations can be mapped to the latent space using the encoder $f_{\mathrm{enc}}(x; \psi)$. As the encoder is a parametric mapping, it can map previously unseen data from the same underlying data distribution as the training data into the latent space.

### Componentwise likelihood-based boosting

In the following, we describe generalized likelihood-based boosting with componentwise linear learners (comp$L_2$Boost)[43,44] (see also refs. 45,78,79). While e.g., in ref. 43 the procedure is described in a more general setting including several versions of generalized linear models, we consider only the case of i.i.d. homoscedastic normally distributed responses, such that the link function corresponds to the identity function. As a result, the version stated here is equivalent to gradient boosting with componentwise linear learners using the squared-error loss[51,80], and to forward stagewise linear regression[49].

The comp$L_2$Boost algorithm is a stagewise variable selection approach that updates only one of the regression coefficients in each iteration. Consequently, boosting results in a sparse estimate of the coefficient vector and

can produce stable estimates even if the number of variables exceeds the number of observations by far. This makes boosting highly useful and successful for high-dimensional omics data, such as from scRNA-seq.

Consider some pairs of observations $(\boldsymbol{x}_i, y_i)_{i=1,\ldots,n}$ of covariate vectors $\boldsymbol{x}_i \in \mathbb{R}^p$, defining the matrix $\boldsymbol{X} := (\boldsymbol{x}^{(1)}, \ldots, \boldsymbol{x}^{(p)}) := (\boldsymbol{x}_1, \ldots, \boldsymbol{x}_n)^\top \in \mathbb{R}^{n \times p}$ and a continuous response vector $\boldsymbol{y} = (y_1, \ldots, y_n)^\top \in \mathbb{R}^n$. The responses $y_i$ are assumed to be realizations of independent homoscedastic random variables $Y_i \sim \mathcal{N}((\boldsymbol{x}_i)^\top \boldsymbol{\beta}, \sigma^2)$ for a fixed $\sigma^2 > 0$ and an unknown coefficient vector $\boldsymbol{\beta} \in \mathbb{R}^p$, which is to be estimated. In the following, we assume that the covariate vectors $\boldsymbol{x}^{(j)}, j = 1, \ldots, p$ are standardized, i.e., $\hat{\mu}_{\boldsymbol{x}^{(j)}} = 0$ and $\sum_{i=1}^n x_{ij}^2 = n - 1$ using the corrected standard deviation estimator.

In the beginning of the algorithm, regression coefficients $\hat{\boldsymbol{\beta}}^{(0)} := (0, \ldots, 0)^\top$ are initialized at 0 and we define the initial offset $\hat{\boldsymbol{\eta}}^{(0)} := (\hat{\eta}_1^{(0)}, \ldots, \hat{\eta}_n^{(0)})^\top = (0, \ldots, 0)^\top$. For a fixed step size $\varepsilon \in (0, 1)$ and a fixed number of boosting steps $M \in \mathbb{N}$, the updates for the regression coefficients in every step $m = 1, \ldots, M$ are computed as follows:

1. For each component $j = 1, \ldots, p$, define the linear predictors

$$\eta_{i,j}^{(m)} = \hat{\eta}_i^{(m-1)} + \gamma_j^{(m)} x_{ij}, i = 1, \ldots, n,$$

for which we want to determine the index $j^*$ resulting in the best fit at the current iteration.

2. For $j = 1, \ldots, p$, compute the maximum likelihood estimator $\hat{\gamma}_j^{(m)}$ of the (partial) log-likelihood

$$l(\gamma_j^{(m)}) := \log\left(\prod_{i=1}^n p(y_i | (\boldsymbol{x}_i, j))\right) = -\frac{n}{2} \log(2\pi\sigma^2)$$
$$- \frac{1}{2\sigma^2} \sum_{i=1}^n \left(\left(y_i - \hat{\eta}_i^{(m-1)}\right) - \gamma_j^{(m)} x_{ij}\right)^2,$$
$$y_i | (\boldsymbol{x}_i, j) \sim \mathcal{N}\left(\eta_{i,j}^{(m)}, \sigma^2\right). \tag{2}$$

Assuming that at least one entry of each covariate vector differs from zero and since only the second summand of the right side of Eq. (2) depends on $\gamma_j^{(m)}$, the maximizer for each index $j = 1, \ldots, p$ can be derived by calculating the unique root of the derivative

$$\frac{\partial l(\gamma_j^{(m)})}{\partial \gamma_j^{(m)}} = \frac{1}{\sigma^2}(\boldsymbol{x}^{(j)})^\top \left(\left(\boldsymbol{y} - \hat{\boldsymbol{\eta}}^{(m-1)}\right) - \gamma_j^{(m)} \cdot \boldsymbol{x}^{(j)}\right),$$
$$\hat{\gamma}_j^{(m)} = \frac{(\boldsymbol{x}^{(j)})^\top (\boldsymbol{y} - \hat{\boldsymbol{\eta}}^{(m-1)})}{(\boldsymbol{x}^{(j)})^\top \boldsymbol{x}^{(j)}}.$$

The $j$th solution equals the score function $S(\gamma_j^{(m)}) = \frac{\partial l(\gamma_j^{(m)})}{\partial \gamma_j^{(m)}}$ of the log-likelihood divided by the fisher information $I(\gamma_j^{(m)}) = -\frac{\partial^2 l(\gamma_j^{(m)})}{\partial (\gamma_j^{(m)})^2}$ both evaluated at $\gamma_j^{(m)} = 0$. Hence, $\hat{\gamma}_j^m$ can be considered as the result of one fisher scoring step with a starting point at zero[43].

3. Out of all the computed maximum likelihood estimators in the current iteration, determine the one $\hat{\gamma}_{j^*}^{(m)}$ with the highest score

$$j^* = \arg\max_{j=1,\ldots,p} \frac{\left(S(\gamma_j^{(m)})|_{\gamma_j^{(m)}=0}\right)^2}{I(\gamma_j^{(m)})|_{\gamma_j^{(m)}=0}}$$
$$= \arg\max_{j=1,\ldots,p} \left((\boldsymbol{x}^{(j)})^\top (\boldsymbol{y} - \hat{\boldsymbol{\eta}}^{(m-1)})\right)^2.$$

Note that the denominator is constant for standardized covariate vectors and can thus be neglected as it is independent of $j$.

4. Compute the updated coefficient vector by

$$\hat{\boldsymbol{\beta}}^{(m)} = \begin{cases} \hat{\beta}_j^{(m-1)} + \varepsilon \hat{\gamma}_j^{(m)} & , j = j^* \\ \hat{\beta}_j^{(m-1)} & , \text{else}. \end{cases}$$

and define the new offset $\hat{\boldsymbol{\eta}}^{(m)} = \hat{\boldsymbol{\eta}}^{(m-1)} + \varepsilon \hat{\gamma}_{j^*}^{(m)} \cdot \boldsymbol{x}^{(j)}$.

The sparsity level of the final coefficient vector is directly regulated by the number of boosting steps where for $M \in \mathbb{N}$ boosting steps, at most $M$ coefficients can become nonzero. Additionally, the step size $\varepsilon$ influences the sparsity level, where either making too large steps or to small steps increases the sparsity of the coefficient vector.

A pseudocode for the version of the comp$L_2$Boost algorithm that we utilize in the optimization process of the BAE (see "Core optimization algorithm of the BAE") for standardized covariate vectors and a mean centered response vector is stated in Supplementary Algorithm 1.

### Core optimization algorithm of the BAE

A BAE is an autoencoder with a specific architecture, where comp$L_2$Boost is integrated into the gradient-based optimization scheme for the training. While the decoder of a BAE can be chosen arbitrarily, the encoder consists of only a linear layer

$$BAE : \Theta \mapsto \mathbb{R}^p,$$
$$BAE(\boldsymbol{x}; (\boldsymbol{B}, \boldsymbol{\theta})) := f_{\text{dec}}(f_{\text{enc}}(\boldsymbol{x}; \boldsymbol{B}); \boldsymbol{\theta}),$$
$$f_{\text{enc}}(\boldsymbol{x}; \boldsymbol{B}) := \rho_{(\cdot)}^{\text{enc}}(\boldsymbol{B}^\top \boldsymbol{x}),$$
$$\rho^{\text{enc}}(x) := \text{id}(x),$$

where $\boldsymbol{B}$ denotes the transposed encoder weight matrix. Note that extensions to more complex encoder architectures are possible, as discussed in "Discussion".

Starting with a zero-initialized encoder weight matrix and randomly initialized decoder parameters, the optimization of the BAE in each training epoch is performed in an alternating manner by optimizing the encoder weights via successive application of the comp$L_2$Boost algorithm to each latent dimension and optimization of the decoder parameters using a gradient-based optimization algorithm such as stochastic gradient descent or Adam. To integrate boosting as a supervised approach into an unsupervised optimization process, we define pseudo responses to update the linear model coefficients, corresponding to the rows of the encoder weight matrix.

Consider observations $\boldsymbol{X} \in \mathbb{R}^{n \times p}$ with standardized covariate vectors and a BAE with $d \ll p$ latent dimensions. Due to standardization, we work with normality assumptions and thus choose the mean squared error as the reconstruction loss:

$$L_{\text{rec}}(\boldsymbol{x}, \hat{\boldsymbol{x}}) := \frac{1}{p} \sum_{j=1}^p (x_j - \hat{x}_j)^2.$$

In an arbitrary training epoch $k \in \mathbb{N}$, let $\boldsymbol{B} = (\boldsymbol{\beta}^{(1)}, \ldots, \boldsymbol{\beta}^{(d)}) \in \mathbb{R}^{p \times d}$ denote the current transposed encoder weight matrix, where each column vector $\boldsymbol{\beta}^{(l)}$ represents the current regression coefficients and $\boldsymbol{Z} = (\boldsymbol{z}^{(1)}, \ldots, \boldsymbol{z}^{(d)}) := \boldsymbol{X}\boldsymbol{B}$ represents the latent representation, consisting of the current prediction vectors $\boldsymbol{z}^{(l)}$, for each latent dimension $l = 1, \ldots, d$. The first step for updating the encoder weights is to compute the negative gradients of the training objective with respect to the prediction vectors $\boldsymbol{z}^{(l)}$ as pseudo

responses, defining

$$\boldsymbol{g}^{(l)} := -\nabla_{\boldsymbol{z}^{(l)}} \frac{1}{n} \sum_{i=1}^{n} L_{\mathrm{rec}}(\boldsymbol{x}_i, f_{\mathrm{dec}}(\boldsymbol{z}_i; \boldsymbol{\theta})), l = 1, \ldots, d.$$

The negative gradients represent the directions in which the predictions have to be adjusted to minimize the training objective the most. We standardize the pseudo responses to fulfill the required normality assumption for the comp$L_2$Boost from "Componentwise likelihood-based boosting", and to omit an intercept term

$$\widehat{\boldsymbol{g}}^{(l)} = (\widehat{g}_1^{(l)}, \ldots, \widehat{g}_n^{(l)})^{\top}, \widehat{g}_i^{(l)} := \frac{g_i^{(l)} - \widehat{\mu}_{\boldsymbol{g}^{(l)}}}{\widehat{\sigma}_{\boldsymbol{g}}^{(l)}}, i = 1, \ldots, n.$$

Note that after standardization, the pseudo responses still define descent directions for the training objective. Subsequently, updates for the regression coefficients, i.e., the weights of the encoder, are obtained via successively applying the comp$L_2$Boost algorithm for only one boosting step with a step size $\varepsilon \in (0, 1)$ to the pairs of observations and pseudo responses $(\boldsymbol{x}_i, \widehat{g}_i^{(l)})_{i=1}^{n}$

$$\boldsymbol{\beta}^{(l)} = \mathrm{comp}L_2\mathrm{Boost}(\boldsymbol{\beta}^{(l)}, \boldsymbol{X}, \widehat{\boldsymbol{g}}^{(l)}; \varepsilon, \mathrm{steps} = 1), l = 1, \ldots, d.$$

Subsequently, the decoder parameters are updated by performing one step of a gradient-based optimization scheme.

After training, the latent representation can be computed by feeding the data to the trained encoder $f_{\mathrm{enc}}$, parameterized by the sparse encoder weight matrix. Each nonzero weight corresponds to a link between a latent dimension and one of the input variables. At the same time, its absolute value is an indicator of the importance of that variable.

Supplementary Algorithm 2 provides a pseudocode of the alternating core BAE optimization algorithm. We refer to this alternating optimization procedure as optimization in alternating mode. Although described in an alternating manner here, the gradients of the training objective w.r.t. $\boldsymbol{z}^{(1)}, \ldots, \boldsymbol{z}^{(d)}$ and the decoder parameters could be computed simultaneously on the same training batches. This would allow for parallelization of the subsequent update computations using different optimization criteria.

Alternatively, encoder and decoder can be trained jointly, which simplifies training by using an end-to-end instead of a two-step strategy, and allows for adding further layers to the encoder for capturing more complex patterns. This requires to differentiate through the boosting step in each epoch to obtain gradients of the reconstruction loss that respect the encoder parameter updates, which can be realized efficiently using differentiable programming[81] and a flexible automatic differentiation framework[53] (see "Differentiable programming for joint optimization").

Note that the activation functions need to be selected such that their derivatives do not vanish at zero, which happens, e.g., for standard implementations of the popular ReLU function. Otherwise the pseudo responses vanish and the encoder weight matrix stays zero during the entire training process.

Applying comp$L_2$Boost for only one boosting step to each latent dimension results in a coarse approximation of the current pseudo responses in each training epoch, where only one element of the associated row vector of the encoder weight matrix is updated. However, applying the algorithm for only one step is sufficient, as the fit is carried out as part of an iterative optimization process.

In general, by training a BAE as described above, with $d$ latent dimensions for a number of $N$ training epochs, at least $d$ elements but at most $d \cdot N$ elements of the encoder weight matrix become nonzero during the training. Consequently, the sparsity level of the encoder weight matrix is highly influenced by the number of training epochs and boosting steps. For reaching a desired sparsity level, we suggest limiting the maximum number of nonzero coefficients per dimension by training for a fixed number of epochs while adjusting the step size for the boosting component, regulating the number of nonzero coefficients. We believe that introducing

regularization for the decoder parameters and the coefficients of the boosting component (see e.g., in refs. 45,70) could further stabilize the variable selection when training for a large amount of epochs and prevent it from overfitting.

To illustrate the optimization process of the BAE and verify its effectiveness in representing distinct cell groups and identifying characteristic sparse gene sets in each latent dimension via modification of the selection criterion, we provide a simple simulation design in Supplementary Note 8 and a detailed analysis in Supplementary Note 9 and Supplementary Figs. 12 and 13. A comparison to potential alternative approaches using regularization techniques is provided in Supplementary Note 10 and Supplementary Figs. 14–19.

## Disentanglement constraint for structuring latent representations

So far, we have explained the BAE optimization using unconstrained gradients as pseudo responses. To learn disentangled latent dimensions, we constrain the pseudo responses for the boosting in the encoder optimization. Consider the same setting as described in "Core optimization algorithm of the BAE for updating the encoder weight matrix". For each latent dimension $l = 1, \ldots, d$, instead of taking the standardized negative gradient $\widehat{\boldsymbol{g}}^{(l)}$ as the pseudo responses, only the fraction of the gradient that is currently not explainable by the other latent dimensions $k \in \{k = 1, \ldots, d | k \neq l\}$ is used as the constrained pseudo responses. Therefore, we first define by

$$\mathcal{I} := \{l\} \cup \{k = 1, \ldots, d | \boldsymbol{\beta}^{(k)} = \boldsymbol{0}_{\mathbb{R}^p}\}$$

the set of all indices of latent dimensions with corresponding all-zero encoder weights together with the current index $l$. Note that zero-weight vectors only occur in the first training epoch. The matrix $\boldsymbol{Z}^{(-\mathcal{I})}$ then consists of only the columns of $\boldsymbol{Z}$ whose indices do not appear in $\mathcal{I}$. The exclusion of zero columns is important for avoiding technical invertibility problems in the following computations. Next, we predict the best linear fit of $\boldsymbol{Z}^{(-\mathcal{I})}$ to the pseudo responses $\widehat{\boldsymbol{g}}^{(l)}$ by first computing the ordinary linear least squares estimator

$$\boldsymbol{\alpha}^* := \underset{\boldsymbol{\alpha} \in \mathbb{R}^{d - |\mathcal{I}|}}{\arg\min} \frac{1}{2} \| \widehat{\boldsymbol{g}}^{(l)} - \boldsymbol{Z}^{(-\mathcal{I})}\boldsymbol{\alpha} \|_2^2$$

$$= ((\boldsymbol{Z}^{(-\mathcal{I})})^{\top} \boldsymbol{Z}^{(-\mathcal{I})})^{-1} (\boldsymbol{Z}^{(-\mathcal{I})})^{\top} \widehat{\boldsymbol{g}}^{(l)},$$

and subsequently calculating the residual vector

$$\boldsymbol{y}^{(l)} := \widehat{\boldsymbol{g}}^{(l)} - \boldsymbol{Z}^{(-\mathcal{I})}\boldsymbol{\alpha}^*.$$

The constrained pseudo responses for the boosting are then defined as the standardized vector

$$\widehat{\boldsymbol{y}}^{(l)} = (\widehat{y}_1^{(l)}, \ldots, \widehat{y}_n^{(l)})^{\top}, \widehat{y}_i^{(l)} := \frac{y_i^{(l)} - \widehat{\mu}_{\boldsymbol{y}^{(l)}}}{\widehat{\sigma}_{\boldsymbol{y}^{(l)}}}, i = 1, \ldots, n,$$

representing the fraction of the original pseudo responses $\widehat{\boldsymbol{g}}^{(l)}$ that is currently not explained by other latent dimensions $k \in \{k = 1, \ldots, d | k \neq l\}$.

In the special case of updating $\boldsymbol{\beta}^{(1)}$ during the first training epoch, where no variables have been selected yet, $\mathcal{I} = \{1, \ldots, d\}$. In that case, we neglect the least squares estimation-step and directly set $\widehat{\boldsymbol{y}}^{(1)} = \widehat{\boldsymbol{g}}^{(1)}$.

A pseudocode for the disentanglement constraint that can be included in the core optimization algorithm for adapting the pseudo response of each latent dimension in each training epoch is stated in Supplementary Algorithm 3.

## Differentiable programming for joint optimization

For better communication between the encoder and decoder parameter updates, we provide an alternative optimization procedure for the BAE.

While in the optimization process described in "Core optimization algorithm of the BAE" updates for the encoder- and decoder parameters are performed in an alternating way, they can also be optimized simultaneously using differentiable programming[52,53,81], a paradigm for flexibly coupling optimization of diverse model components. For an overview from a statistical perspective, see ref. 82. Specifically, we incorporate the boosting part of the optimization as part of a recursively defined joint objective $L_{joint}(X; \theta)$

$$\widehat{B}_\theta = \left( \text{comp} L_2 \text{Boost}\left( \beta^{(l)}, X, \widehat{g}_\theta^{(l)}; \varepsilon, \text{steps} = 1 \right) \right)_{l=1,\ldots,d},$$

$$\widehat{Z}_\theta = X\widehat{B}_\theta,$$

$$L_{joint}(X; \theta) = \frac{1}{n} \sum_{i=1}^{n} L_{rec}(x_i, f_{dec}(\widehat{Z}_\theta; \theta)_i).$$

Hence, updates for the decoder parameters respect changes in the encoder parameters. Thereby, the $\theta$ subscript indicates the dependence on the current state of the decoder parameters. This procedure is facilitated by differentiable programming, which allows for jointly optimizing the neural network and boosting components by providing gradients of a joint loss function with respect to the parameters of all components via automatic differentiation. In particular, we use the Zygote.jl automatic differentiation framework[53], which allows for differentiating through almost arbitrary code without user refactoring.

By each call to the joint objective, when optimizing the decoder parameters in one training epoch via an iterative gradient-based optimization algorithm, boosting is performed in the forward pass, hence, the encoder weights are updated in the forward pass.

The joint version of the objective allows to adapt our alternating optimization algorithm to a stochastic framework, meaning that the joint objective can also be applied to randomly sampled mini-batches consisting of $m \leq n$ of the observation vectors. This enables the simultaneous optimization of the encoder and decoder parameters using the same mini-batch. Thus, the boosting part of the BAE optimization process acts on minibatches, which is related to the stochastic gradient boosting approach described in ref. 83.

Note that using the joint objective function for the optimization leads to a slight increase in training duration since the calculation of the reconstruction loss also incorporates the boosting step. Also, depending on the implementation of the mini-batch procedure, each training epoch could consist of updating all model parameters iteratively, using a partition of the whole dataset into distinct mini-batches of size $m$. Consequently, the boosting component could be applied several times per epoch, which affects the sparsity level of the encoder weight matrix.

We refer to this joint optimization procedure as optimization in jointLoss mode.

## Adaptation for time series

As an additional use case, we illustrate an adaption of the BAE for handling time series data. Therefore, consider a $T$-tuple consisting of snapshots of observation data at different time points $(X^{(1)}, \ldots, X^{(T)})$. E.g., these could be scRNA-seq datasets from the same tissue sequenced at different time points. The matrices might contain different numbers of observations/cells, i.e., $X^{(i)} \in \mathbb{R}^{n_i \times p}$, for $n_1, \ldots, n_T \in \mathbb{N}, t = 1, \ldots, T$.

We assume that the data reflects a developmental process of cells across measurement time points. For example, cells differentiate into different (sub-)types, and each process is characterized by a small gene set that gradually changes over time by sequential up- and downregulation of genes along the developmental trajectory. For each developmental process, we aim to capture the changes in the associated gene set in linked latent dimensions, one for each time point in an adapted version of the BAE, called timeBAE.

More precisely, the timeBAE comprises one BAE for each time point. The models are trained successively with the disentanglement constraint (see sections "Core optimization algorithm of the BAE" and "Disentanglement constraint for structuring latent representations") for a fixed number of epochs on data from the corresponding time point using a pre-

training strategy. At each time point, the encoder weight matrix from the previous time point is used for initializing the current encoder weight matrix. Once the complete training process is done, we concatenate all the encoder weight matrices and end up with a new stacked encoder weight matrix with $d \cdot N$ latent dimensions. For each $k = 1, \ldots, d$, dimensions $k + (t - 1)d$ for $t = 1 \ldots T$ then represent the same developmental process across time. This requires some prior knowledge about the expected number $d \in \mathbb{N}$ of developmental processes of cell groups.

For a formal description, consider $T$ BAEs with linear encoders $f_{enc}^{(T)}(x; B^{(T)}) := (B^{(T)})^\top x$, where at the beginning of the training, $B^{(0)} := 0_{\mathbb{R}^{p \times d}}$. Each BAE is trained with the data from the respective time point. First, decoder parameters are initialized randomly. We denote the updated matrix obtained after training at time $t$ by $B^{(t)}$, and the next initialization of the encoder weight matrix at time $t + 1$ by $\widetilde{B}^{(t+1)}$. We set $\widetilde{B}^{(1)} := B^{(0)}$. Specifically, for the following time points we set

$$\widetilde{B}^{(t+1)} := B^{(t)} - \widetilde{B}^{(t)}, t = 1, \ldots, T.$$

Thus, changes in previous encoder weights are incorporated as prior knowledge guiding the model to select genes corresponding to the same developmental program in the same dimension across time points. After training, the overall transposed encoder weight matrix is defined as $B := (\widetilde{B}^{(2)}, \ldots, \widetilde{B}^{(T+1)}) \in \mathbb{R}^{p \times d \cdot T}$, and the joint encoder as $f_{enc}(x; B) := B^\top x$.

A reordered matrix $B_{perm}$, where the linked latent dimensions across time are grouped, can be constructed by reorganizing the columns of $B$ as follows:

$$B_{perm} = \left( \widetilde{B}_{(:,1)}^{(2)} \quad \cdots \quad \widetilde{B}_{(:,1)}^{(T+1)} \quad \cdots \quad \widetilde{B}_{(:,d)}^{(2)} \quad \cdots \quad \widetilde{B}_{(:,d)}^{(T+1)} \right) \in \mathbb{R}^{p \times d \cdot T},$$

where $\widetilde{B}_{(:,j)}^{(t)}$ represents the $j$th column of $\widetilde{B}^{(t)}$, for $t \in \{2, \ldots, T + 1\}$ and $j \in \{1, \ldots, d\}$.

Note that this approach is compatible with both training in alternating and in jointLoss mode.

## Determining correspondence of latent patterns to cell types

In situations where cell labels of $k$ categories, such as cell types, are already available, the learned patterns in different BAE latent dimensions can be checked for overlap with the label patterns. For this purpose, a quantile thresholding strategy is used to check the top positive and negative representation values of each latent dimension for overlap with the cell type patterns. Let $Z \in \mathbb{R}^{n \times d}$ represent the $d$-dimensional BAE latent representation of $n$ cells. For $j = 1, \ldots, d$ we first compute the $p \in (0, 1)$ quantile $q_+^{(j)}$ and the $(1 - p)$ quantile $q_-^{(j)}$ given the representation vector $z^{(j)}$ in dimension $j$. Next, we determine the indices of cells with a representation value above $q_+^{(j)}$ and check the different label proportions $p_{+,l}^{(j)} \in [0, 1], l = 1, \ldots, k$, e.g., cell type proportions, and do the same for cells with a representation value below $q_-^{(j)}$. The label that best matches the learned positive patterns in the latent dimension $j$ is

$$\underset{l \in \{1, \ldots, k\}}{\arg \max} \quad p_{+,l}^{(j)}.$$

The label that best matches the learned negative patterns can be computed in a similar way. For the BAE analysis on the cortical mouse data, we set $p = 0.9$.

## Identification of top selected genes per latent dimension

In contrast to standard iterative gradient-based parameter optimization, which starts from a random initialization of the parameters, we presented an optimization approach that starts from a zero initialization and allows the iterative selection of the parameters to be updated. Iteratively updating the BAE encoder weights using componentwise boosting results in a sparse

encoder weight matrix where many of the weights are truly zero. However, the nonzero weights in the encoder weight matrix link variables, i.e., genes, with different latent dimensions, and the magnitude of their values reflects the effect on the representation in the dimensions. Taken together, the two most important pieces of information that can be derived from the encoder weight matrix are (1) knowledge of which genes have been selected (nonzero weights), and (2) how strongly the genes are associated with the latent dimensions. The latter is especially important when training for a large number of training epochs, where weights are selected and updated in a forward selection manner. To identify the top genes, i.e., the most influential genes for the different latent dimensions, we suggest examining per individual latent dimension, the scatter plots of the encoder weights sorted in descending order by their absolute value. In addition, a change-point criterion can be used to identify the genes with a nonzero weight per latent dimension that have the greatest influence on the representation. The change-point genes for a latent dimension can be identified by first sorting the nonzero absolute values of the corresponding encoder weights in descending order. Then, pairwise differences are computed for the neighboring elements in the sorted vector. The maximum value of the differences indicates where the change point is, i.e., where the order of magnitude of the absolute values of the weights decreases the most. For example, if the maximum value of the differences is the third element, then the first three genes are considered to be the change-point genes, i.e., the top genes.

### Label predictions of unseen data
Given that the cells used to train the model are labeled, label predictions for unseen cells can be performed. First, the unseen cells are mapped to the latent space using the trained encoder of the BAE. Labels for these cells are then predicted by identifying the $k$ nearest neighbors in the low-dimensional latent space, using a distance metric such as Euclidean distance. The most frequent cell type among these $k$ neighbors is assigned as the predicted label for each unseen cell. For the analysis of the cortical mouse data in "The BAE identifies cell types and corresponding marker genes in cortical neurons", we set $k = 10$.

### Gene selection stability analysis
To investigate the robustness in terms of gene selection stability on the cortical mouse data, we repeatedly trained a BAE under the same training conditions as described in "Statistics and reproducibility", but using different initializations for the decoder parameters. We determined the number of times each gene was selected across 30 runs. For each run, we examined whether the weights corresponding to the gene in at least one latent dimension were nonzero. If so, the gene was considered selected for that run, contributing a count of 1. The total selection count for each gene was then calculated as the sum of its selections across all 30 runs.

### Robustness analysis of structuring latent representations
The robustness of the BAE was assessed in its ability to learn well-structured latent representations under more challenging conditions, such as increased noise levels or weaker signals. To achieve this, we leveraged the gene selection stability analysis results for the BAE (see "Gene selection stability analysis", Supplementary Note 4, Supplementary Fig. 6, and Supplementary Table 3) and systematically replaced the top $q\%$ of the most frequently selected genes in the cortical mouse dataset (1525 cells and 1500 highly variable genes) with noise genes. Noise genes were defined as randomly selected genes that were expressed in at least one cell and distinct from the set of 1500 highly variable genes. We trained the BAE under the same training conditions outlined in "Statistics and reproducibility" for different percentages $q$ and visually compared the outcomes. The results are presented in Supplementary Note 5, Supplementary Fig. 7, and Supplementary Table 4.

### Clustering analysis of BAE and PCA based representations of cortical neurons
We trained a BAE with 10 latent dimensions using 20 different decoder parameter initializations in alternating mode with the disentanglement

constraint on cortical neurons stemming from the primary visual cortex of mice. After training, we computed the corresponding 20 latent representations of the data. On the other hand, we computed the PCA representation of the data where we only kept the first ten principal components equaling the number of latent dimensions of the BAE. Since PCA is a deterministic approach, we performed Leiden clustering for 20 seeds, resulting in 20 different PCA clusterings. In contrast, we performed Leiden clustering once for each BAE representation.

Subsequently, we evaluated and compared the clustering results using two popular metrics for cluster analysis, the silhouette score and the adjusted Rand index (e.g., see refs. [84–86]). The silhouette score is a general quality measure for a single clustering, which is defined as the mean of silhouette coefficients of the clustering. For each observation, the corresponding silhouette coefficient measures how well the observation fits into its cluster compared to others. The silhouette coefficient takes values in $[-1, 1]$. A value of one corresponds to an observation fitting well to its assigned cluster compared to other clusters, and a value of $-1$ means that the observation fits better to other clusters compared to its own. The adjusted Rand index is a version of the Rand index that is corrected for random chance measuring the similarity of two clusterings. A value of one means that both clusterings are identical, whereas a value of zero corresponds to random labeling. We used the scikit-learn implementation for both clustering metrics.

We report the median silhouette score of the 20 clusterings for each method together with the interquartile ranges of silhouette scores, as well as the median adjusted Rand index, for quantitatively comparing the clustering results (see "The BAE identifies cell types and corresponding marker genes in cortical neurons").

### Model comparison for gene selection
We compared the BAE to two other sparse unsupervised variable selection approaches for gene selection: (1) sparsePCA[40] and (2) single-cell Projective Non-negative Matrix Factorization (scPNMF)[65]. Like the BAE, both approaches aim to establish sparse connections from the gene space to different dimensions in a low-dimensional space. Brief descriptions of these methods are given below and detailed explanations are available in the respective references[40,65].

**sparsePCA**. sparsePCA[40] is a modification of the original PCA formulation that additionally constrains the resulting loading matrix to be sparse. Let $X \in \mathbb{R}^{n \times p}$ denote the data matrix with $n$ observations, e.g., cells and $p$ variables, e.g., genes. The optimization problem solved can be formulated as a PCA problem (dictionary learning[87]) with an $\ell_1$ penalty on the $d \in \mathbb{N}$ components:

$$\underset{U,V}{\arg\min} \quad \| X - UV \|_F + \alpha \| V \|_1$$
$$\text{subject to} \quad \| u^{(l)} \|_2 \leq 1, \quad \forall l \in \{1, \ldots, d\}$$

where $\| \cdot \|_F$ denotes the Frobenius norm, $\| \cdot \|_2$ is the Euclidean norm, and $\| \cdot \|_1$ is the sum of the absolute values of all matrix entries. The sparsity level can be adjusted through the hyperparameter $\alpha$. Increased values of $\alpha$ lead to an increased sparsity level of the resulting loading matrix $V \in \mathbb{R}^{d \times p}$. For our analysis, we used the scikit-learn (v1.4.0) implementation of sparsePCA which is based on ref. [87].

**scPNMF**. scPNMF[65] is an extension of the Projected Non-negative Matrix Factorization (PNMF) algorithm[88,89] that is constrained to learn a sparse loading matrix consisting of only positive elements, specifically tailored for scRNA-seq data analysis.

The algorithm takes as input a log-transformed count matrix $X \in \mathbb{R}_{\geq 0}^{n \times p}$, with $n$ cells and $p$ genes. Given a number of dimensions $d \in \mathbb{N}$, the goal of PNMF is to identify a representation of the data in a $d$-dimensional space in which the dimensions are defined as nonnegative, sparse, and mutually exclusive linear combinations of the $p$ genes[65]. To achieve this,

PNMF solves the following optimization problem:

$$\arg\min_{W \in \mathbb{R}_{\geq 0}^{p \times d}} \| X - XWW^\top \|_F.$$

Here, $W$ is a matrix whose entries are in addition constrained to be nonnegative. The optimization problem is initialized with a nonnegative loading matrix and solved using an iterative optimization algorithm[65]. During each update iteration, elements in $W$ that fall below a specified threshold are set to zero.

A second important step in the scPNMF workflow is post-hoc basis selection of the resulting loading matrix, where dimensions are filtered whose corresponding components do not reflect a biological pattern or are correlated with cell library sizes[65]. However, since post-hoc basis selection could in principle be performed also for the BAE, we excluded it from our comparison and concentrate on the comparison of the outputted loading matrix, i.e., encoder weight matrix.

For the model comparison, we used the same dataset for both the BAE and sparsePCA, comprising 1,525 neural cells from the cortical mouse data. The data was standardized prior to parameter optimization, as detailed in "Data preprocessing". For scPNMF, log-transformed normalized count values with a pseudo-count of one were used instead. All approaches were restricted to selecting genes from the set of the 1500 most highly variable genes. We ran sparsePCA and scPNMF with their default hyperparameter settings, but set the number of components, i.e., latent dimensions, to 10. The approaches were compared in terms of gene selection, specifically the sparsity of the sparsePCA and scPNMF loading matrices and the BAE encoder weight matrix. Consistent with the gene selection stability analysis (see "Gene selection stability analysis"), we trained the BAE 30 times under identical training conditions but with different initializations of the decoder parameters. Similarly, sparsePCA and scPNMF were run 30 times with varying seed parameters. A gene was considered selected in a run if it had at least one nonzero element across the 10 components in the resulting BAE encoder weight matrix or loading matrices. The results are presented in Supplementary Note 6 and Supplementary Table 5.

## Simulation of scRNA-seq-like data with time structure

To simulate scRNA-seq-like data with time structure representing a developmental process of three different cell groups across three time points, we followed the simulation design presented in ref. 35 (see also Supplementary Note 8, Supplementary Fig. 12, and Supplementary Table 7). We generated one binary count matrix for every time point representing a snapshot of the current state of the three cell groups. To enable a more realistic scenario, the data at each time point consists of a different number of cells and $p = 80$ genes. Specifically, a cell taking the value 1 for some gene indicates a gene expression level exceeding a certain threshold for that cell, whereas the value 0 indicates an expression level below that threshold. The data at each time point is modeled by i.i.d. random variables $(X_{i,j} \sim \mathrm{Ber}(p_{i,j}))_{i=1,\ldots,n,j=1,\ldots,p}$, where $p_{i,j}$ denotes the probability of a value of 1 for a gene $j$ in a cell $i$. We set $p_{i,j}$ to 0.6 for highly expressed genes (HEGs) and to 0.1 for lowly expressed genes (LEGs). To model a smooth transition between two successive cell stages across subsequent time points, we include overlaps between the sets of HEGs from two subsequent time points. Table 1 comprises the details of the gene expression patterns and overlap patterns for the count matrices at different time points. In total there are 54 HEGs for the different cell stages and 26 genes that are lowly expressed in all cell stages

across all time points that are added as non-informative genes to make the variable selection task more challenging and realistic. For simplicity, we strictly excluded intersections between the sets of HEGs of distinct cell groups. An exemplary visualization of simulated time series scRNA-seq data can be found in Fig. 4a.

## Gene selection comparison for the timeBAE

In a setting where both temporal information and cluster identities are available, we can compare the sets of selected genes of the timeBAE to the ones obtained by fitting linear models given the cluster and time point information. The process involves the following steps:

The trimmed embryoid body dataset is first divided into subsets corresponding to different time points ($X^{(1)}, \ldots, X^{(T)}$) in the same way as for training the timeBAE. Next, for each time point $t$ and cluster $c$, binary response variables are defined as follows:

$$y_i^{(c,t)} = \begin{cases} 1, & \text{if observation } i \text{ at time } t \text{ belongs to cluster } c \\ 0, & \text{otherwise} \end{cases},$$

leading to response vectors:

$$y^{(c,t)} \in \{0, 1\}^{n_t},$$

where $n_t$ is the number of cells measured at time $t$, and $t \in \{1, \ldots, T\}$, $c \in \{1, \ldots, C\}$. We fit multivariate linear regression models without intercept terms using standardized predictors and responses at each time point:

$$\left(X_{\mathrm{st}}^{(t)}, y_{\mathrm{st}}^{(c,t)}\right), \quad c = 1, \ldots, C, t = 1, \ldots, T.$$

Despite the binary nature of the response, linear models are employed to focus on identifying significant coefficients representing characteristic genes rather than predictive accuracy. To identify the genes most characteristic of a cluster at a given time point, we test for each coefficient if it significantly differs from zero via a two-sided t-test. P-values are adjusted for multiple testing using the Bonferroni correction for each fitted regression model. Only genes corresponding to coefficients that were significantly different from zero were counted as selected (coefficients with adjusted p-values below the significance level $\alpha = 0.05$).

Inspecting the UMAP plots colored by the timeBAE latent dimensions, each set of linked timeBAE latent dimensions is visually assigned to the corresponding cluster. For each set of linked latent dimensions, we next compute the intersection of the selected genes for each latent dimension and the set of genes selected by the corresponding linear model. The results are reported in Supplementary Table 6.

## Statistics and reproducibility

All code to run the model and reproduce all analysis results and figures is publicly available at https://github.com/NiklasBrunn/BoostingAutoencoder, including documentation and a tutorial in a Jupyter notebook using the Julia programming language, illustrating the functionality of the approach in a simulation study (see Supplementary Note 8, Supplementary Table 7, and Supplementary Fig. 12).

The BAE and its modification for analyzing time series data are implemented in the Julia programming language[90], v1.6.7, using the packages BenchmarkTools (v1.5.0), CSV (v0.10.11), Clustering (v0.15.7), ColorSchemes (v3.24.0), DataFrames (v1.3.5), Distances (v0.10.11),

## Table 1 | Highly expressed gene patterns in simulated time series scRNA-seq data

| | Cell group 1 (*n* = 310) | Cell group 2 (*n* = 298) | Cell group 3 (*n* = 306) |
|---|---|---|---|
| HEGs (time point 1) | 1–8 | 21–28 | 41–48 |
| HEGs (time point 2) | 6–13 | 26–33 | 46–53 |
| HEGs (time point 3) | 11–18 | 31–38 | 51–58 |

Distributions (v0.25.100), Flux (v0.12.1), GZip (v0.5.2), MultivariateStats (v0.10.3), Plots (v1.24.3), ProgressMeter (v1.9.0), StatsBase (v0.33.21), UMAP (v0.1.10), VegaLite (v2.6.0), XLSX (v0.10.0). Scripts for downloading and preprocessing the embryoid body data and parts of the preprocessing of the cortical mouse data are implemented in Python v3.10.11. We added a YAML file to our GitHub repository containing the information about the used packages and corresponding versions. Users can create a conda environment for running the Python code from the YAML file.

In all our experiments, the encoder of any BAE is defined by

$$f_{enc}(\boldsymbol{x}; \boldsymbol{B}) = \boldsymbol{B}^\top \boldsymbol{x},$$

where the matrix $\boldsymbol{B} \in \mathbb{R}^{p \times d}$ denotes the transposed encoder weight matrix and we used 2-layered decoders

$$f_{dec}(\boldsymbol{z}; (\boldsymbol{W}^{(1)}, \boldsymbol{b}^{(1)}, \boldsymbol{W}^{(2)}, \boldsymbol{b}^{(2)})) = \boldsymbol{W}^{(2)} \tanh_{(\cdot)}(\boldsymbol{W}^{(1)} \boldsymbol{z} + \boldsymbol{b}^{(1)}) + \boldsymbol{b}^{(2)},$$

where $\boldsymbol{W}^{(1)} \in \mathbb{R}^{p \times d}$, $\boldsymbol{b}^{(1)} \in \mathbb{R}^p$, $\boldsymbol{W}^{(2)} \in \mathbb{R}^{p \times p}$ and $\boldsymbol{b}^{(2)} \in \mathbb{R}^p$ denote the trainable weight matrices and bias vectors.

The number of units in the input and output layers corresponds to the number of variables in the datasets (i.e., $p = 1500$ for the cortical mouse data analysis and subgroup analysis, $p = 80$ in the time series simulation, $p = 263$ on the embryoid body dataset, and $p = 50$ in the stair-like simulation design from Supplementary Note 8). The number of latent dimensions was selected according to a rough estimate of the number of groups present or expected in the datasets. Specifically, we used $d = 10$ latent dimensions for the cortical mouse data comprising cells of 15 neuron cell types, $d = 4$ for the subgroup analysis of *Sst*-neurons, $d = 3$ in the time series simulation where we simulated three groups of developing cells, $d = 4$ for the embryoid body data where we obtained three clusters from Leiden clustering with a resolution parameter of 0.025, and $d = 10$ in the stair-like simulation design where we simulated ten groups of cells.

In the boosting part of the optimization, we set the number of boosting steps to $M = 1$ and the step size to $\varepsilon = 0.01$ for all experiments. Specifically, we applied BAEs using the disentanglement constraint in alternating mode to obtain the presented results for the embryoid body data, the clustering analysis on cortical mouse data and the stair-like simulation design, and in jointLoss mode for obtaining the results for the remaining datasets. We chose $m = 500$ as the common batchsize for optimizing encoder and decoder parameters for the cortical mouse data, $m = 95$ for application on the *Sst*-neurons, $m = 198$ for the simulated time series scRNA-seq data, $m = 1500$ as the batchsize for decoder parameter optimization for the embryoid body data, and $m = 800$ as the batchsize for decoder parameter optimization for the stair-like simulation design. In every scenario, we used the Adam optimizer[62] for updating the decoder parameters, where the learning rate was set to $\nu = 0.01$. The number of training epochs was set to 25 for obtaining the here presented results on the cortical mouse data, 20 for the *Sst*-neurons, 9 for training the timeBAE with data at each time point of the simulated scRNA-seq data, 15 for training with data at each time point of the embryoid body data, and 15 for the stair-like simulation design.

## Computation time and memory
A key focus in the present manuscript is the applicability of the BAE to small data scenarios, e.g., for helping to identify and characterize rare cell types. We have thus not specifically investigated on scalability in the current

implementation. In Table 2, we present the computation time and total allocated memory for the optimization process in the main experiments. All experiments were conducted on a MacBook Pro (2023) with an Apple M2 Max chip (12-core CPU, 38-core GPU) and 96 GB of RAM. The required computation times and memory were measured using the @btime macro from Julia's BenchmarkTools package. The reported times and memory represent the minimum execution time and total allocated memory across multiple runs, ensuring robust performance evaluation by minimizing noise.

## Data preprocessing
For the adult mouse cortical scRNA-seq data[61], we downloaded the expression data consisting of the measured gene expression levels of 15,119 genes in 1679 single cells, the gene names, and cell type annotations from GEO under accession number GSE71585. Next we removed the samples which have been flagged by Tasic et al.[61] as being of bad quality. As the quantification of aligned reads per gene has been conducted with RSEM[91], decimal numbers might show up occasionally in the estimated counts per gene. Next, we normalized the data using the DESeq2 normalization for sequencing depth[92] to adjust for fluctuations in the number of aligned reads per sample, affecting each gene's estimated expression level. Afterward, we log-transformed the normalized counts with an added pseudo-count of one. Genes were removed if count values did not exceed ten in more than 20 cells. Our analysis focused on the subset of cortical neurons based on the cell type annotation provided by Tasic et al.[61]. We removed genes expressed in less than ten cells for the remaining cells. Further, we reduced the set of genes to 1500 highly variable genes using the built-in scanpy (v1.9.8) function for highly variable gene selection with the flavor variable set to 'seurat', including marker genes of neural cell types (Fig. 3 and Supplementary Fig. 12 in ref. 61) and neurotransmitter receptor genes (Supplementary Fig. 15 in ref. 61) that were previously identified in ref. 61 (Fig. 3 in ref. 61). The resulting count matrix consists of 1525 neurons and 1500 highly variable genes, including 90 of the 180 neural marker and receptor genes identified by Tasic et al.[61].

In the subgroup analysis of cells highly expressing the marker *Sst*, we first selected only the *Sst*-neurons. Subsequently, we removed genes expressed in less than ten cells, followed by highly variable gene selection using the built-in scanpy (v1.9.8) function for highly variable gene selection with the flavor variable set to "seurat". The resulting count matrix consists of 202 *Sst*-neurons with gene counts of 1500 highly variable genes.

The embryoid body data comprises gene expression levels of embryonic cells generated over a 27 day time course, stored in five different count matrices depending on measurement time. Especially, cells sequenced at days 0–3 were labeled as time point 1, cells from day 6 to 9 as time point 2, cells from day 12 to 15 as time point 3, cells from day 18 to 21 as time point 4, and cells from day 24 to 27 as time point 5. We excluded cells from time point 1 since their expression patterns are mostly homogeneous. For the remaining cells, we followed and adapted the preprocessing steps from ref. 66. First, we filtered cells by library size to remove doublets and empty droplets, where for each time point cells below the lower 20th percentile and above the upper 20th percentile were excluded. Next, we filtered genes expressed in less than ten cells, separately for each time point. We further excluded apoptotic cells expressing a large total amount of mitochondrial genes The threshold was set to 400. Subsequently, we normalized the counts for each cell by dividing each count value by the total number of counts across genes. Next, we re-scaled the normalized counts by a factor of 10,000,

**Table 2 | Required computation time and memory for the main experiments**

|  | Cortical mouse data | Embryoid body data | Simulated data BAE (Supplementary Note 8) | Simulated data timeBAE |
|---|---|---|---|---|
| Model | BAE | timeBAE | BAE | timeBAE |
| Mode | jointLoss | alternating | alternating | jointLoss |
| Computation time (in seconds) | 254.14 | 21.17 | 0.12 | 0.68 |
| Allocated memory (in GB) | 119.04 | 11.42 | 0.21 | 0.71 |

followed by a log-transformation with a pseudo-count of one. For the timeBAE analysis, we fused the data matrices from time points 3 and 4, creating the new time point 34 for those cells, and subset the data to cells that can be grouped in large clusters. Therefore, we performed PCA and chose the first 50 principal components to apply the Leiden clustering algorithm with resolution parameters of 0.025 and 0.25. Under observation of the clustering results with the resolution parameter 0.25 in a 2D UMAP plot of cells, we then removed clusters that were separated from the larger groups. Next, we selected genes that were differentially expressed between different clusters of the clustering with resolution parameter 0.025 using the built-in scanpy (v1.9.8) function for differentially expressed gene detection with the method variable set to "wilcoxon". In addition, we selected genes differentially expressed between the adapted time points for each individual cluster. After preprocessing and filtering, we ended up with 11,627 cells and a combined set of 262 differentially expressed genes.

Since the BAE approach requires observation data in a standardized form, z-scores of the gene vectors were computed for each of the simulated and real-world scRNA-seq datasets. We used the log-transformed counts to standardize the mouse cortical scRNA-seq data and the embryoid body scRNA-seq data. For the simulated scRNA-seq datasets, we re-scaled the noise variables after standardization by multiplying them with a scalar factor to adjust the range of the values to that of the other gene vectors. The scaling factor for the simulated scRNA-seq data with time structure was set to 0.5, and for the stair-like scRNA-seq data for the analysis of the BAE functionality in Supplementary Note 9 to $\frac{2}{3}$.

All scripts for loading and preprocessing the different datasets are available in our GitHub repository https://github.com/NiklasBrunn/BoostingAutoencoder.

### Reporting summary

Further information on research design is available in the Nature Portfolio Reporting Summary linked to this article.

### Data availability

The scRNA-seq datasets used in this study are publicly available. The adult mouse cortical dataset from ref. 61 can be accessed from the Gene expression Omnibus (GEO) at https://www.ncbi.nlm.nih.gov/geo/ under the accession number GSE71585 (GSE71585_RefSeq_counts.csv.gz). The embryoid body scRNA-seq data from ref. 66 is available from the Mendeley Data repository at https://data.mendeley.com/datasets/v6n743h5ng/1 with the identifier doi: 10.17632/v6n743h5ng.1[93].

### Code availability

Code for the simulated data, for preprocessing and the proposed algorithm is available at https://github.com/NiklasBrunn/BoostingAutoencoder, including a tutorial Jupyter notebook to illustrate the approach and scripts to reproduce all figures and analysis results shown in the manuscript. Further details on the implementation can be found in "Statistics and reproducibility".

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

## Acknowledgements
This work was funded by the Deutsche Forschungsgemeinschaft (DFG, German Research Foundation)—Project-ID 322977937—GRK 2344 (H.B., M.H., N.B., and T.V.) and Project-ID 499552394—SFB 1597 (H.B. and M.H.). The authors thank Moritz Hess for help in preprocessing the adult mouse cortical data and Laia Canal Guitart for help in preprocessing the embryoid body dataset.

## Author contributions
H.B. proposed the original idea and supervised the project. M.H. and N.B. implemented the approach and designed the computational experiments. N.B. performed the computational analyses. M.H. and N.B. wrote the manuscript. T.V. aided in the biological interpretation and in revising the manuscript. All authors revised and approved the final manuscript.

## Funding

## Competing interests
The authors declare no competing interests.
