## [Transparent Peer Review file · Communications Biology]

Infusing structural assumptions into dimensionality reduction for single-cell RNA sequencing data to identify small gene sets

Corresponding Author: Mr Niklas Brunn

This manuscript has been previously reviewed at another journal. This document only contains information relating to versions considered at Communications Biology.

Version 0:

Reviewer comments:

Reviewer #1

(Remarks to the Author)

The manuscript outlines a Boosting Autoencoder (BAE) approach to conduct dimensionality reduction and facilitate feature selection in scRNA-seq data. It addresses a significant challenge in the analysis of scRNA-seq data, which is often tackled using less accurate, highly randomized methods such as UMAP and t-SNE. The autoencoder (AE) category of models show promise to address these challenges, and AE models have previously been applied to scRNA-seq data for similar purposes. Here, the authors propose another variant of AEs, BAEs, to improve performance in clustering, highlight important gene groups, and extract trends in developmental cycles. The authors provide the code for their method in a well-organized Github repository with helpful guides and dependencies. Overall, the manuscript addresses an important problem with a novel solution. There are areas where improvements can be made.

Major Issues:

1-Clarity and Quantification - section 2.2- Page 7:

The statement "Each dimension captures a group of cells that mostly corresponds to a specific annotated cell type." Requires clarification. What does "mostly" mean in this context? Reporting a mean count and a confidence interval might be helpful. The authors continue: "Inspection of the corresponding selected genes shows that the top genes for each dimension are among the listed neural marker genes of the respective cell types". The terms "top genes" requires quantification. How many genes are considered as "top genes"; what is the percentage in each case. Can recall/accuracy rates, and their standard deviation, be calculated across multiple runs? Providing these numbers would facilitate the comparison with alternative methods, which is another missing point of discussion here. This will quantify the contributions of this novel method. how do other methods perform in highlighting important genes in comparison to the BAE.

2- Comparison with other methods – section 2.2 – Page 8:

There is a comparison against PCAs in dimensionality reduction efficacy in clustering, where both methods' outputs are used as input for UMAP. No significant differences is found between the BAE and PCA (p-value is not reported). What justifies the use of BAE, if it shows no improvement over PCA and is not an "end-to-end" solution (considering UMAP and Leiden were run on the outputs of both). Focusing on tasks where BAE shows improvements would clarify the manuscript message (at least in the main text).

3- Variance Criteria – section 2.3 – page 10:

In the statement "we select the 1,500 most highly variable genes on the subpopulation", the criteria for selecting the most variable genes is not specified. Adding a figure illustrating these criteria in the supplementary material would be beneficial. Additionally, in the same section/page (also in Figure 3a,b), it is important to clarify that the PCA representation used to generated the PCA-based UMAP was also trained using only the n=202 samples that the BAE was trained on. This paragraph could also benefit from numerical reporting, such as p-values and confidence intervals to compare the 2 models.

4- Measurement of method's power – section 2.5 – page 14:

There is no measurement of method's power or its comparison to alternative methods. In the comparison with the regression

analysis on the same page, there are no numbers reported (significance of the overlap). Additionally, if the regression analysis is the gold standard, why not use the regression instead of BAE?

Was there anything done to independently confirm the subtypes of Sst-expressing neurons in the subgroup analysis correspond to a biological process and not data artifacts?

5- Power Analysis:

The claim about the power of clustering in small groups (“Even if a very small group is captured by such a clustering, the statistical power to infer marker genes is often limited due to the small sample size”) could benefit from analysis that demonstrates the proposed method provides higher power compared to alternatives.

Recommendation:

The authors address several challenges in this paper. While it is commendable, it results in the manuscripts message becoming somewhat vague. A more focused approach will improve clarity and highlight the main contributions of BAE. Focusing on a limited number of claims providing robust empirical evidence will achieve this. The most exciting aspects of this paper are scenarios where the BAE pipeline excels, such as the time-series analysis for the development cycle, and specification of subtypes.

Minor Issues:

1. Terminology:

The authors use the expression “dimension reduction” which is technically not wrong. However, “dimensionality reduction” is the more commonly used term and is recommended.

2. Methods section:

The methods section is thorough and considers the audience needs by going into details. However, clarity could be improved. There might be a mistake on page 18, section 5.2, first paragraph, in the definition of dimensionality of $W(i)$. It seems n_i and n_{i-1} should be swapped. Or notation should be updated later to indicate that matrix is transposed.

Reviewer #2

(Remarks to the Author)

In this paper, a boosting autoencoder (BAE) model is introduced for interpretable dimensionality reduction for scRNA-seq data, with the possibility to constrain the model to certain desirable features such as disentanglement. The model is built from 2 components: a single linear layer that learns through component-wise boosting serving as the encoder, and a standard neural network decoder built from stacking linear layers and (non-linear) activation functions. The model is unsupervised, although a supervised variant with similar performance is discussed and presented. The unsupervised functionality is achieved by training the boosting encoder using the negative gradients of the reconstruction loss with respect to the latents, which is an intuitive approach for selecting the most important genes for dimensionality reduction.

The main advantage of the BAE model is that it is more interpretable than typical autoencoders, since each gene is directly linked to a latent dimension through the encoder weight matrix, facilitating an analysis such as selecting the top weights. The design has multiple benefits, for example marker genes are selected during training, i.e. in a single step as opposed to standard approaches where this step is done separately after training. Also, the algorithm encourages sparse weight matrices and sparse latent embeddings, which helps in finding small, representative gene sets.

I think the algorithm is interesting and should be published as an alternative to standard approaches. The combination of the boosting encoder and traditional decoder could motivate further research into explainable neural networks. Here, my main concern would be in terms of scalability and usage for modern datasets. While I am not a fan of the “bigger is always better” mentality in deep learning, most modern scRNA-seq studies tend to be on the scale of hundreds of thousands of cells (e.g. cell atlases). The authors are aware of this potential limitation, which mostly comes from the single linear encoder layer. The success of neural networks can often (though not always) be attributed to stacking several non-linear layers, and it is not clear if the BAE can be as successful on datasets that are several orders of magnitude larger than presented here. The success of BAE on very small datasets is admirable, though I am not sure that it is a common application nowadays, except for some fine-grained selection where it could be the tool of choice. Still, the authors proposed some possible extensions for a more complex encoder which could mitigate some of these limitations. As such, this paper could be seen as a first step in this direction.

Another possible shortcoming would be the comparison to other methods. In the current article, there are several comparisons with a standard autoencoder and a few variations with additional loss terms, e.g. for disentanglement. As a sidenote, this is currently in Supplementary Section 7 but could be more emphasised in the main paper. The BAE outperformed or was simpler to use than these variations which is encouraging. At the same time, the comparison does not consider some methods like the variational autoencoder which is generally regarded as more powerful/useful than the plain autoencoder. On a similar note, although multiple real-world and artificial datasets are used in the evaluation, we do not have a measure of how “difficult” they are. Other works have looked at datasets that are known to be difficult to cluster, e.g. schistosome stem cells in the “Self-assembling manifolds” and “CellVGAE” papers.

I will also point out a few minor issues that could in my opinion be improved:

1. In the description of the algorithm, it can be difficult to follow the meaning of terms like “component”, “dimension”, “row”, “gene”, etc. and which refers to which. It would be useful to clearly state how the data is structured (e.g. cells X genes), and

in the algorithms whether the indices range over genes, latent dimensions, etc. These could be followed in the current version, mostly, but it can be confusing sometimes.

2. On a related note, there are extensive presentations of the approach/algorithms both in the main text, methods, and first figure. I think that the presentation could be tidier and more focused, as the same things seem to be explained in slightly different ways in different places, which can be confusing when first reading through the paper.
3. I also think that the term “criterion” encountered throughout the paper is a bit too abstract, and terms like “outcome-based criterion” are unfamiliar for a deep learning audience. Perhaps this could be better contextualized, or an alternative term could be used.
4. For Figure 5b, it is not clear how the 3 dimensions are chosen from the 12.
5. There are 2 places in the paper stating: “results not shown”. These should be shown in the Supplementary Materials, especially the predictions on test set cell types.
6. An idea for a further experiment could be to quantify how the marker gene sets change/evolve over training epochs (maybe there isn't a notable change but at least a comment on this could be useful)
7. It might be interesting to experiment with dropout as another regularisation technique, for example in the decoder and maybe even in the encoder.
8. It would be useful to have some data on the required time and memory to train a BAE model, for both the “alternating” and “JointLoss” strategies (I expect these to be quite low but it is always good to have an objective measure)
9. A family of (V)AE models used for scRNA-seq data relies on graph variational autoencoders. It would be interesting if the authors can comment on possible extensions of the model for the graph domain and if they anticipate any benefits and/or problems.

Overall, I think that the method is scientifically interesting and that it provides some convenience and performance benefits compared to traditional approaches. The application to time-series data is another useful capability of the model, and other constraints or settings could be designed in the future. The hybrid statistical-deep learning approach could also motivate more research into this direction. The BAE model might exhibit limitations for large and diverse datasets, and thus the immediate applicability of the model might be impacted. The content so far should be enough for an initial publication, with the possibility to address the scalability issue in the future. The work is reproducible and the code for the model is hosted on an open-source platform.

Reviewer #3

(Remarks to the Author)

The authors proposed a new structure of autoencoder, boosting autoencoder (BAE), which combines autoencoder for dimensionality reduction and boosting for sparsity. They applied BAE on scRNA-seq data and showed that this method provides simultaneous dimensionality reduction and gene selection. Although the method is clearly written, I have several questions about comparing it to other methods and some results.

1. The method essentially uses a linear encoder. There are several existing methods for sparse linear dimensionality reduction. For example, we have sparse PCA and single cell Projective Matrix Factorization (scPNMF, Bioinformatics 2021). Such methods also provide a sparse representation of the data and are much simpler compared to the BAE framework. The authors should perform some benchmarking on these methods to further show the superiority of BAE.
2. For the timeBAE, “the encoder weights for a given time point are initialized with the learned encoder weights from the previous time point.” I am curious how much difference between the final encoders from this initialization compared to those from the random initialization. This will further show the importance of using these sequential encoder weights.
3. For Figure 3b, the pattern looks quite weak. If my understanding is correct, the authors are showing that in cluster 2, the BAE dimensions are related to time points. However, this trend is not very clear; cell types and time points are tangled together. I wonder if the authors can improve this result or have a better explanation if this is my misunderstanding.

Version 1:

Reviewer comments:

Reviewer #2

(Remarks to the Author)

The authors have comprehensively addressed my concerns and I think that the manuscript is improved compared to the original submission. As such, I think the paper can be published in its current form.

Reviewer #3

(Remarks to the Author)

The authors have addressed all my previous questions. I do not have more questions for this manuscript.

Point-by-point response to reviewer comments

Manuscript Title: Infusing structural assumptions into dimension reduction for single-cell RNA sequencing data to identify small gene sets

Submitted to: Nature – Communications Biology

Reviewer #1:

The manuscript outlines a Boosting Autoencoder (BAE) approach to conduct dimensionality reduction and facilitate variable selection in scRNA-seq data. It addresses a significant challenge in the analysis of scRNA-seq data, which is often tackled using less accurate, highly randomized methods such as UMAP and t-SNE. The autoencoder (AE) category of models show promise to address these challenges, and AE models have previously been applied to scRNA-seq data for similar purposes. Here, the authors propose another variant of AEs, BAEs, to improve performance in clustering, highlight important gene groups, and extract trends in developmental cycles. The authors provide the code for their method in a well-organized Github repository with helpful guides and dependencies. Overall, the manuscript addresses an important problem with a novel solution. There are areas where improvements can be made.

Response: Thank you for your assessment of our manuscript and for believing in the importance of our proposed BAE approach. We have comprehensively addressed and clarified all issues in the revised version of the manuscript.

Comment 1 (Clarity and Quantification - section 2.2 - Page 7): The statement “*Each dimension captures a group of cells that **mostly** corresponds to a specific annotated cell type.*” Requires clarification. What does “mostly” mean in this context? Reporting a mean count and a confidence interval might be helpful. The authors continue: “*Inspection of the corresponding selected genes shows that the top genes for each dimension are among the listed neural marker genes of the respective cell types*”. The terms “top genes” requires quantification. How many genes are considered as “top genes”; what is the percentage in each case? Can recall/accuracy rates and their standard deviation be calculated across multiple runs? Providing these numbers would facilitate the comparison with alternative methods which is another missing point of discussion here. This will quantify the contributions of this novel method. How do other methods perform in highlighting important genes in comparison to the BAE.

Response: Thank you for the comments on Section 2.2 of our original manuscript. To clarify the statement on the cell groups, we now provide a quantile thresholding strategy, which we describe in a new Methods Section 5.8 (line 797-809 in the main text), to identify the most frequent cell type for the cells above the 90% quantile and the cells below the 10% quantile separately for each latent dimension. The percentages of the best matching types are shown in Supplementary Figure S2 and are consistent with the previously described results. In addition, we now have updated the part of the main text mentioned in your comment (line 201-203 in the main text).

Also, we recognize the benefit of providing more guidance in defining the most important genes per dimension. So far, we have relied on examining our gene scatterplots, which show the normalized coefficients of the k BAE-selected genes with the highest absolute values in descending order. In addition, we now added a change-point criterion to identify for each latent dimension the gene for which the largest jump, i.e. change-point, occurs in the sorted absolute weight values in descending order and keep only the genes up to the identified one. The genes before the change-point contribute the most to the representation values in the corresponding latent dimension. We added a description of the procedure in the new Methods Section 5.9 (line 810-832 in the main text). In addition, we added a table comprising the change-point genes for the analysis of the cortical mouse data in Supplementary Section S3 (Table 2), a table summarizing information about the number of selected genes per latent dimension and change-point genes per latent dimension (Table 1), and updated the respective part of the main text (line 203-207 in the main text).

Furthermore, we added a gene selection stability analysis for the BAE on the cortical mouse data and examined the genes that were most frequently selected. Specifically, we trained a BAE for 30 times under identical training conditions, but with different decoder parameter initializations. We determined the number of times each gene was selected across 30 runs. For each run, we examined whether the weights corresponding to the gene in at least one latent dimension were nonzero. If so, the gene was considered selected for that run, contributing a count of 1. The total selection count for each gene was then calculated as the sum of its selections across all 30 runs. We summarized the results of the gene selection stability analysis together with an examination of the genes that were selected in at least 24 runs (80% of the 30 runs) in a bar plot (Supplementary Figure S6) and a table (Supplementary Table 3) in Supplementary S6. In addition, we added a new Methods Section 5.11 (line 840-847 in the main text) describing the gene stability analysis and updated the main text in Section 2.2 (line 215-223 in the main text).

Finally, in response to your important suggestion regarding a comparison with other methods for highlighting important genes, we included an analysis comparing the BAE with sparsePCA and sparsePNMF (as recommended by another reviewer) in terms of gene selection (line 268-272 in the main text). A description of this analysis has been added to the new Methods Section 5.14 (line 882-930 in the main text). The results are presented in Supplementary Section S9 and summarized in Supplementary Table 5. In brief, the BAE selected the fewest genes and had the lowest number of nonzero elements in the encoder weight matrix, reflecting minimal gene-to-latent dimension connections.

Comment 2 (Comparison with other methods – section 2.2 – Page 8): There is a comparison against PCAs in dimensionality reduction efficacy in clustering where both methods' outputs are used as input for UMPA. No significant differences is found between the BAE and PCA (p-value is not reported). What justifies the use of BAE if it shows no improvement over PCA and is not an “end-to-end” solution (considering UMAP and Leiden were run on the outputs of both)? Focusing on tasks where BAE shows improvements would clarify the manuscript message (at least in the main text).

Response: Thank you for pointing out the lack of clarity in this explanation. In the comparison to PCA, our aim was not to show superiority, but rather to exemplarily show that using the BAE is not inferior to PCA, i.e., does not result in a lack of performance in standard downstream analysis tasks, such as clustering and UMAP visualizations. Thus, we wanted to show that whenever a researcher would use a standard PCA representation in scRNA-seq analysis, they could alternatively use a BAE representation without losing performance, but gaining the method's unique advantages, i.e., the selection of a sparse, interpretable gene set, and the flexibility in encoding different structural constraints. To clarify this in the manuscript, we have now re-written the explanation in Section 2.2 (line 252-267 in the main text), and additionally included this point into the discussion (line 461-464 in the main text). We agree that the BAE is not an “end-to-end” solution for obtaining a clustered UMAP visualization. Rather, it is an end-to-end approach for identifying

groups of cells in a low-dimensional representation together with sparse sets of characterizing genes, which would correspond to separate analysis steps in a standard scRNA-seq workflow.

We appreciate your suggestion to put a stronger focus on tasks where the BAE shows improvements. In our view, the key advantages of using the BAE are its ability to provide a sparse set of genes associated with each latent dimension, which facilitate biological interpretation of the patterns captured in the different dimensions, and its flexibility to adapt the gene selection criterion to different experimental setups, as exemplified by the time series analysis. To highlight these strengths, we have put more focus on the gene selection component in our revision by providing a change-point approach for computationally identifying top genes and a gene selection stability analysis, as explained in our response to your Comment 1, and enhanced the time series application by improved explanations and additional analysis to showcase the flexible adaptability of the BAE to different setups (see below in the response to your Comment 4).

Comment 3 (Variance Criteria – section 2.3 – page 10): In the statement “we select the 1500 most highly variable genes on the subpopulation” the criteria for selecting the most variable genes is not specified. Adding a figure illustrating these criteria in the supplementary material would be beneficial. Additionally in the same section/page (also in Figure 3ab) it is important to clarify that the PCA representation used to generate the PCA-based UMAP was also trained using only the n=202 samples that the BAE was trained on. This paragraph could also benefit from numerical reporting such as p-values and confidence intervals to compare the 2 models.

Response: We agree that the selection of highly variable genes mentioned in Section 2.3 needs clarification. In our revised version, we have adapted the description of the preprocessing steps for all datasets in Methods Section 5.19 (line 1038-1042, line 1045-1047, and 1067-1072 in the main text) to specify the criteria used to select highly variable genes and differentially expressed genes. Specifically, we used the Python package scanpy (v1.9.8) to preprocess the data and the scanpy function `sc.pp.highly_variable_genes()` with `flavor='seurat'` for the mouse cells to select highly variable genes and the scanpy function `sc.tl.rank_genes_groups()` with `method='wilcoxon'` for the embryoid body cells. We have also added a note in this section that all scripts for loading and preprocessing the datasets can be found in our GitHub repository <https://github.com/NiklasBrunn/BoostingAutoencoder> (line 1081-1082 in the main text).

In addition, we added the specification that PCA was also applied only to the subset of n=202 Sst-expressing neurons in Section 2.3 (line 285-287 in the main text).

Comment 4 (Measurement of method's power – section 2.5 – page 14): There is no measurement of method's power or its comparison to alternative methods. In the comparison with the regression analysis on the same page, there are no numbers reported (significance of the overlap). Additionally if the regression analysis is the gold standard, why not use the regression instead of BAE? Was there anything done to independently confirm the subtypes of Sst-expressing neurons in the subgroup analysis correspond to a biological process and not data artifacts?

Response: Thank you for raising these important points. We respond to your concerns regarding the method's power in detail below in our response to your following Comment 5. Here, we focus on your comments on the regression analysis and the justification for using the timeBAE.

In the revised manuscript, we now report the quantitative details of the comparison between the timeBAE and the regression analysis in Supplementary Section S12, specifically the percentage of overlap between the top genes identified by the timeBAE latent dimensions and the significant genes identified by the regression models. These results confirm that the genes identified by the timeBAE are consistent with those detected by the regression analysis.

We agree that we have not stated the purpose of the regression clearly enough. We had not intended

the regression as a gold standard, but rather we used the regression to provide a reference annotation in a setting without ground truth information on the lineages and their markers. In particular, the regression approach is only applicable if cluster labels are available, and if each cluster corresponds to a distinct developmental trajectory. We therefore deliberately simplified the dataset to create such a setting. The timeBAE does not have access to the cluster label, i.e., it is at a disadvantage compared to the regression. The regression is therefore not intended as a performance competitor, but serving only to validate that timeBAE can identify meaningful structure, and is only applicable because we deliberately created a suitable simplification of the dataset.

We have now clarified this in more detail in the revised text in Section 2.5 (line 399-417 in the main text) and in the discussion (line 465-473 in the main text). We emphasize in particular that our setting was chosen deliberately for evaluation purposes, and that in more realistic, complex scenarios, the regression approach would no longer be applicable, while the timeBAE remains effective. As the evaluation setting corresponds to a somewhat artificial simplification, we would be hesitant to put a stronger emphasis on the comparison beyond the numerical reporting and additional clarification.

As mentioned in the manuscript, we examined the genes selected by BAE in the subgroup analysis of *Sst*-expressing neurons with the highest corresponding absolute weight values per dimension using our gene scatter plots and compared them to genes identified as different *Sst* subgroup markers in [61] (Figure 3 and Supplementary Figure S12). We have adjusted the corresponding part in Section 2.3 for clarity (line 292-294 in the main text). For the purposes of our manuscript, our subgroup analysis of *Sst*-expressing neurons focused exclusively on the comparison of the genes selected by the BAE model with the marker genes listed in [61], i.e. no further experiments or analyses were performed to confirm that the subgroups correspond to a biological process and not to data artifacts.

Comment 5 (Power Analysis): The claim about the power of clustering in small groups (“Even if a very small group is captured by such a clustering, the statistical power to infer marker genes is often limited due to the small sample size”) could benefit from analysis that demonstrates the proposed method provides higher power compared to alternatives.

Response: Thank you for this comment. We agree that the statement you have mentioned was misleading. Rather than commenting on the power of clustering, we wanted to point out that if a cluster consists of only very few cells, the identification of marker genes to annotate the cluster, using standard differential expression testing, is complicated by the small sample size. This problem is supported by findings from the cited research articles (e.g., Su et al. 2020, Märtens et al. 2022). The BAE is an end-to-end approach that does not perform clustering, but selects a sparse set of genes for the pattern captured in each latent dimension, which can correspond to a group of cells that would also have been captured by a clustering but does not have to. In contrast, the standard approach to annotate groups of cells with marker genes would be clustering, followed by separate differential gene expression testing. These tests requires pre-defined cluster labels, which are not computed or used in the BAE framework, such that a formal power comparison of the gene identification is not possible. Rather than such a formal power analysis, we wanted to highlight that the end-to-end approach of the BAE, which does not rely on differential expression tests based on fixed cluster labels, can be beneficial particularly in settings with small groups, as we have shown in the manuscript. To avoid confusion, we have now rephrased the misleading statement in the introduction (line 99-101 in the main text).

Comment 6 (Recommendation): The authors address several challenges in this paper. While it is commendable it results in the manuscript’s message becoming somewhat vague. A more focused approach will improve clarity and highlight the main contributions of BAE. Focusing on a limited number of claims

providing robust empirical evidence will achieve this. The most exciting aspects of this paper are scenarios where the BAE pipeline excels such as the time-series analysis for the development cycle and specification of subtypes.

Response: Thank you for this valuable feedback, we greatly appreciate your recommendation. In the revised manuscript, we now put a stronger focus on the method's main strengths. In particular, we investigated more in detail the methods' ability to select small, biologically meaningful gene sets, which we consider a core strength, by proposing a change-point strategy for automatically identifying the most informative genes, and sensitivity analyses, specifically by re-calculating informative genes across different runs and investigating robustness to removal of different percentages of these top genes (for the robustness analysis see: line 223-228, line 848-857 in the main text, and Supplementary Section S7). We present this in Section 2.2 and provide additional details in the Methods Section and in the Supplementary Material (Section S3/6/7), as also explained in detail above in our response to your Comment 1.

We have also aimed at providing further robust evidence of the benefits of the BAE pipeline by comparing it to other approaches for sparse dimensionality reduction, specifically sparse PCA and scPNMF, as elaborated in Section 2.2. Further, we now provide more detailed explanations and additional results for the time-series analysis in Section 2.4 (line 361-365 and Supplementary Section S10) and 2.5.

We have chosen the different application scenarios in the manuscript to illustrate the adaptability of the BAE, which we also perceive as one of its core strengths. Yet, we understand your concern that this may lead to vagueness in the main message. We have therefore clarified the method's main strengths in the main text and in the discussion, where we have added a sentence stating the main benefits of the BAE and the reason for the choice of different application settings (line 448-452 in the main text).

Comment 7 (Minor Issues): 1. **Terminology:** The authors use the expression "dimension reduction" which is technically not wrong. However, "dimensionality reduction" is the more commonly used term and is recommended. 2. **Methods section:** The methods section is thorough and considers the audience's needs by going into details. However, clarity could be improved. There might be a mistake on page 18 section 5.2 first paragraph in the definition of dimensionality of $W(i)$. It seems n_i and n_{i-1} should be swapped. Or notation should be updated later to indicate that the matrix is transposed.

Response: We thank the reviewer for pointing out these issues. In the revised version of the manuscript, we exchanged the term dimension reduction with the more frequently used term dimensionality reduction and corrected the dimensionality of $W(i)$ in the Methods Section 5.2 (line 551 in the main text).

Reviewer #2:

In this paper, a boosting autoencoder (BAE) model is introduced for interpretable dimensionality reduction for scRNA-seq data with the possibility to constrain the model to certain desirable variables such as disentanglement. The model is built from 2 components: a single linear layer that learns through component-wise boosting serving as the encoder and a standard neural network decoder built from stacking linear layers and (non-linear) activation functions. The model is unsupervised although a supervised variant with similar performance is discussed and presented. The unsupervised functionality is achieved by training the boosting encoder using the negative gradients of the reconstruction loss with respect to the latents which is an intuitive approach for selecting the most important genes for dimensionality reduction.

The main advantage of the BAE model is that it is more interpretable than typical autoencoders since each gene is directly linked to a latent dimension through the encoder weight matrix facilitating an analysis such as selecting the top weights. The design has multiple benefits for example marker genes are selected

during training i.e. in a single step as opposed to standard approaches where this step is done separately after training. Also the algorithm encourages sparse weight matrices and sparse latent embeddings which helps in finding small representative gene sets.

I think the algorithm is interesting and should be published as an alternative to standard approaches. The combination of the boosting encoder and traditional decoder could motivate further research into explainable neural networks.

Response: We greatly appreciate your evaluation of the paper and that you see potential for further research in the field of explainable AI that is in our case facilitated by the integration of boosting into a neural network architecture. We have comprehensively addressed your remaining concerns in our revised version of the manuscript, and respond to each comment in detail below.

Comment 1: Here my main concern would be in terms of scalability and usage for modern datasets. While I am not a fan of the “bigger is always better” mentality in deep learning, most modern scRNA-seq studies tend to be on the scale of hundreds of thousands of cells (e.g., cell atlases). The authors are aware of this potential limitation which mostly comes from the single linear encoder layer. The success of neural networks can often (though not always) be attributed to stacking several non-linear layers and it is not clear if the BAE can be as successful on datasets that are several orders of magnitude larger than presented here. The success of BAE on very small datasets is admirable though I am not sure that it is a common application nowadays except for some fine-grained selection where it could be the tool of choice. Still, the authors proposed some possible extensions for a more complex encoder which could mitigate some of these limitations. As such, this paper could be seen as a first step in this direction.

Response: Thank you for bringing up the important issue of scalability. We agree that the single encoder linear might be a limitation when applying the BAE on substantially more complex datasets, and now explain possible ways to address this in the discussion. In particular, we have chosen a linear encoder to prioritize interpretability over flexibility, and have designed the approach with a specific focus on small data scenarios, such as characterizing rare cell types, which can be very challenging using standard approaches, but is of high biological relevance. Adding more layers might compromise interpretability due to the loss of direct gene to latent dimension linkage. Nonetheless, we agree that exploring more complex architectures is a valuable direction for future research. In our revision, we have now expanded our previous discussion to also discuss the applicability on larger datasets (line 480-484 in the main text), and have added a paragraph on scalability (line 492-497 in the main text).

While we are confident in principle that the BAE can also be successfully applied on larger datasets, the current implementation does not focus on scalability. To provide more details, we now also report computation time and memory usage in the Methods Section 5.18 (line 1016-1025 in the main text).

Comment 2: Another possible shortcoming would be the comparison to other methods. In the current article, there are several comparisons with a standard autoencoder and a few variations with additional loss terms e.g. for disentanglement. As a sidenote, this is currently in Supplementary Section 7 but could be more emphasized in the main paper. The BAE outperformed or was simpler to use than these variations which is encouraging. At the same time, the comparison does not consider some methods like the variational autoencoder which is generally regarded as more powerful/useful than the plain autoencoder.

Response: In the revised manuscript, we have addressed the perceived lack of model comparisons by including a comparison of the BAE with sparsePCA and scPNMF for gene selection in the main text in Section 2.2 (line 268-272 in the main text). Further details are explained in Methods Section 5.14 (line 882-930 in the main text) and results are provided in Supplementary Section S9, as referenced in the main text. Furthermore, we expanded the model comparison on simulated data to include a VAE. The additional results

are now presented in Supplementary Section S15 and Supplementary Figure S19. As shown, the VAE was not able to learn a sparse encoder matrix, and its latent dimensions exhibited slightly higher correlations compared to the other approaches.

Comment 3: On a similar note although multiple real-world and artificial datasets are used in the evaluation, we do not have a measure of how “difficult” they are. Other works have looked at datasets that are known to be difficult to cluster e.g., schistosome stem cells in the “Self-assembling manifolds” and “CellV-GAE” papers.

Response: Our primary goal was not to showcase the BAE’s superiority in generating well-clustered latent representations. Instead, in the revised manuscript, we emphasize the tasks where the BAE excels. These include learning sparse and structured encoder weight matrices, identifying sparse gene sets linked to disentangled latent dimensions, and demonstrating flexibility in adapting to diverse tasks, such as capturing cell differentiation patterns characterized by sparse gene sets over time.

Additionally, we now perform a robustness analysis of BAE latent representations using different adaptations of the cortical mouse data (Section 2.2 (line 223-228 in the main text) and Methods Section 5.12 (line 848-857 in the main text)). Here, we systematically replaced certain informative genes from the 1500 highly variable genes identified by the BAE (detailed in the newly added gene selection stability analysis (line 215-223 in the main text), described in Methods Section 5.11 (line 840-847 in the main text) and Supplementary Section S6) with noise genes, such that the resulting datasets become more difficult to cluster. The results, presented in Supplementary Section S7, show that the BAE robustly preserved the core neural cell structure even when a significant portion of informative genes was replaced with noise.

I will also point out a few minor issues that could in my opinion be improved:

Comment 4 & 5 & 6: 1. In the description of the algorithm it can be difficult to follow the meaning of terms like “component” “dimension” “row” “gene” etc. and which refers to which. It would be useful to clearly state how the data is structured (e.g., cells X genes) and in the algorithms whether the indices range over genes latent dimensions etc. These could be followed in the current version mostly but it can be confusing sometimes. 2. On a related note there are extensive presentations of the approach/algorithms both in the main text methods and first figure. I think that the presentation could be tidier and more focused as the same things seem to be explained in slightly different ways in different places which can be confusing when first reading through the paper. 3. I also think that the term “criterion” encountered throughout the paper is a bit too abstract and terms like “outcome-based criterion” are unfamiliar for a deep learning audience. Perhaps this could be better contextualized or an alternative term could be used.

Response: We thank the reviewer for these comments, which helps to improve the readability and clarity of our manuscript. Specifically, we have added a statement at the beginning of the Methods Section 5.1 and Section 2.1 clarifying that observations in our applications correspond to cells and variables to genes (line 123-125 and line 536-537 in the main text). We have now largely standardized the use of different terms that refer to the same thing. Additionally, we have streamlined the presentation of the approach in the first figure and the main text, and removed redundancies. We have further removed the term “outcome-based criterion” and have re-phrased or contextualized most other occurrences of the term “criterion” (line 118-120, line 132-133, and line 142-143 in the main text).

Comment 7: 4. For Figure 5b it is not clear how the 3 dimensions are chosen from the 12.

Response: We have included a formal description of the reordering process for the latent dimensions in the final timeBAE encoder weight matrix in Methods Section 5.7 (line 793-796 in the main text). This addition clarifies how the dimensions are systematically reordered and how the visualized dimensions in Figure 5b

are selected from the 12 latent dimensions of the timeBAE encoder weight matrix (see also line 386 in the main text).

Comment 8: 5. There are 2 places in the paper stating: “results not shown”. These should be shown in the Supplementary Materials especially the predictions on test set cell types.

Response: The predictions on the held-out test set are now shown in Supplementary Section S4 (see also line 207-209 and line 833-839 in the main text) and the results of the regression analysis are shown in Supplementary Section S12 (see also line 399-417 and line 950-972 in the main text).

Comment 9: 6. An idea for a further experiment could be to quantify how the marker gene sets change/evolve over training epochs (maybe there isn't a notable change but at least a comment on this could be useful)

Response: We agree that analyzing how the gene selection evolved during training can provide further insight into the approach and its optimization. In the Supplementary Section S15 (Figure S14), we provide heatmaps showing the evolution of encoder weights, which directly correspond to the coefficients of selected genes, for a simulated dataset. For the real datasets, similar plots are not feasible due to the larger number of genes, most of which have a zero coefficient. However, for providing insights, we have added plots showing the dynamics of the encoder weights per individual latent dimension for the application on the cortical mouse data in Supplementary Section S3, Figure S3.

Comment 10: 7. It might be interesting to experiment with dropout as another regularization technique for example in the decoder and maybe even in the encoder.

Response: Thank you for this suggestion. As the BAE is inherently regularized by the boosting component, we have not considered adding further regularizations so far. Yet, we agree that this would be an interesting direction for extending the approach, and have added a corresponding statement about potential regularization techniques to the discussion (line 498-501 in the main text).

Comment 11: 8. It would be useful to have some data on the required time and memory to train a BAE model for both the “alternating” and “JointLoss” strategies (I expect these to be quite low but it is always good to have an objective measure).

Response: Thank you for highlighting the missing information regarding time and memory requirements. In response, we have added a Methods Section (5.18), where we provide a table summarizing the computation time and memory usage for all major experiments (line 1016-1025 in the main text). Additionally, we have included a statement on the computational efficiency of our current implementation in the discussion (line 492-497 in the main text).

Comment 12: 9. A family of (V)AE models used for scRNA-seq data relies on graph variational autoencoders. It would be interesting if the authors can comment on possible extensions of the model for the graph domain and if they anticipate any benefits and/or problems.

Response: Indeed graph variational autoencoders as well as graph attention networks have gained popularity in recent years and we agree that this would be an exciting extension. We have now added a statement about such an extension to graph-based neural network architectures, how it could work and what could be gained to the discussion (line 515-524 in the main text).

Comment 13: Overall I think that the method is scientifically interesting and that it provides some convenience and performance benefits compared to traditional approaches. The application to time-series data is

another useful capability of the model and other constraints or settings could be designed in the future. The hybrid statistical-deep learning approach could also motivate more research into this direction. The BAE model might exhibit limitations for large and diverse datasets and thus the immediate applicability of the model might be impacted. The content so far should be enough for an initial publication with the possibility to address the scalability issue in the future. The work is reproducible and the code for the model is hosted on an open-source platform.

Response: We appreciate your positive feedback and consideration of our work for publication.

Reviewer #3:

The authors proposed a new structure of autoencoder boosting autoencoder (BAE) which combines autoencoder for dimensionality reduction and boosting for sparsity. They applied BAE on scRNA-seq data and showed that this method provides simultaneous dimensionality reduction and gene selection. Although the method is clearly written I have several questions about comparing it to other methods and some results.

Response: We thank the reviewer for reviewing and commenting on our manuscript. Below we have responded to all of the listed concerns in detail.

Comment 1: 1. The method essentially uses a linear encoder. There are several existing methods for sparse linear dimensionality reduction. For example, we have sparse PCA and single-cell Projective Matrix Factorization (scPNMF Bioinformatics 2021). Such methods also provide a sparse representation of the data and are much simpler compared to the BAE framework. The authors should perform some benchmarking on these methods to further show the superiority of BAE.

Response: We appreciate your suggestion to compare the BAE approach with sparse linear projection methods, such as sparsePCA and scPNMF. While both techniques perform unsupervised variable selection, scPNMF is specifically tailored for analyzing scRNA-seq data. We have now added a comparison of the BAE with sparsePCA and scPNMF regarding gene selection (line 268-272 in the main text). A description of this analysis has been added to the new Methods Section 5.14 (line 882-930 in the main text). The results are presented in Supplementary Section S9 and summarized in Supplementary Table 5. In brief, the BAE selected the fewest genes and had the lowest number of nonzero elements in the encoder weight matrix, reflecting minimal gene-to-latent dimension connections.

Although these methods utilize linear mappings to construct sparse loading matrices for disentangled representations and gene selection, we claim that incorporating the boosting mechanism within a neural network-based framework offers enhanced flexibility for model design. For example, our approach can be adapted to respect the temporal structure of data, as elaborated in Section 2.4 of the manuscript. Additionally, future extensions could explore alternative design choices, such as refining the boosting update criterion or integrating components to address tasks like end-to-end clustering, batch integration, or multi-omics data analysis.

Comment 2: 2. For the timeBAE “the encoder weights for a given time point are initialized with the learned encoder weights from the previous time point.” I am curious how much difference between the final encoders from this initialization compared to those from the random initialization. This will further show the importance of using these sequential encoder weights.

Response: Thank you for pointing out the importance of showing the advantage of our initialization strategy compared to other basic initialization strategy. Since initializing the weights at random via frequently used uniform sampling strategies would result in very dense connections of genes to latent dimensions (al-

ready from the beginning of the training process), we instead choose a zero-initialization as one possible competitor. We added a sentence in the main text in Section 2.4, where we investigate this idea by running the timeBAE on the simulated data with our proposed transfer learning strategy for encoder weight initialization at the next timepoint and compare it to a timeBAE version where the encoder weights are initialized as zeros at each new time point (line 361-365 in the main text). The results are presented in Supplementary Section S10. As can be seen, the timeBAE with the zero-initialization was also capable in learning structured and sparse encoder weight patterns. However, this comes with a loss of linkage of latent dimensions across time, which is one of the key advantages of the timeBAE.

Comment 3: 3. For Figure 3b the pattern looks quite weak. If my understanding is correct the authors are showing that in cluster 2 the BAE dimensions are related to time points. However this trend is not very clear; cell types and time points are tangled together. I wonder if the authors can improve this result or have a better explanation if this is my misunderstanding.

Response: We believe you are referring to Figure 5b in the main text, specifically to Cluster 3, which depicts the latent activations of all cells mapped by a set of linked rows of the learned encoder weight matrices across time points in the timeBAE model. It is important to clarify that these clusters should not be interpreted as biological cell types. Instead, we applied broad clustering (Leiden clustering with a low-resolution parameter) to group cells from different time points after filtering and preprocessing. The clustering was designed to group transcriptionally similar cells across time points, with cells from the same time point generally clustering together. The primary objective of this analysis was to demonstrate the timeBAE's ability to link latent dimensions that capture time-related transcriptional patterns associated with specific cell groups (i.e., clusters) using real-world time-resolved scRNA-seq data. We believe that Figure 5b effectively illustrates this capability.

The manuscript outlines a Boosting Autoencoder (BAE) approach to conduct dimensionality reduction and facilitate feature selection in scRNA-seq data. It addresses a significant challenge in the analysis of scRNA-seq data, which is often tackled using less accurate, highly randomized methods such as UMAP and t-SNE. The autoencoder (AE) category of models show promise to address these challenges, and AE models have previously been applied to scRNA-seq data for similar purposes. Here, the authors propose another variant of AEs, BAEs, to improve performance in clustering, highlight important gene groups, and extract trends in developmental cycles. The authors provide the code for their method in a well-organized Github repository with helpful guides and dependencies. Overall, the manuscript addresses an important problem with a novel solution. There are areas where improvements can be made.

Major Issues:

1-Clarity and Quantification - section 2.2- Page 7:

The statement “*Each dimension captures a group of cells that **mostly** corresponds to a specific annotated cell type.*” Requires clarification. What does “mostly” mean in this context? Reporting a mean count and a confidence interval might be helpful. The authors continue: “*Inspection of the corresponding selected genes shows that the top genes for each dimension are among the listed neural marker genes of the respective cell types*”. The terms “top genes” requires quantification. How many genes are considered as “top genes”; what is the percentage in each case. Can recall/accuracy rates, and their standard deviation, be calculated across multiple runs? Providing these numbers would facilitate the comparison with alternative methods, which is another missing point of discussion here. This will quantify the contributions of this novel method. how do other methods perform in highlighting important genes in comparison to the BAE.

2- Comparison with other methods – section 2.2 – Page 8:

There is a comparison against PCAs in dimensionality reduction efficacy in clustering, where both methods’ outputs are used as input for UMAP. No significant differences is found between the BAE and PCA (p-value is not reported). What justifies the use of BAE, if it shows no improvement over PCA and is not an “end-to-end” solution (considering UMAP and Leiden were run on the outputs of both). Focusing on tasks where BAE shows improvements would clarify the manuscript message (at least in the main text).

3- Variance Criteria – section 2.3 – page 10:

In the statement “we select the 1,500 most highly variable genes on the subpopulation”, the criteria for selecting the most variable genes is not specified. Adding a figure illustrating these criteria in the supplementary material would be beneficial.

Additionally, in the same section/page (also in Figure 3a,b), it is important to clarify that the PCA representation used to generate the PCA-based UMAP was also trained using only the n=202 samples that the BAE was trained on. This paragraph could also benefit from numerical reporting, such as p-values and confidence intervals to compare the 2 models.

4- Measurement of method’s power – section 2.5 – page 14:

There is no measurement of method’s power or its comparison to alternative methods. In the comparison with the regression analysis on the same page, there are no numbers

reported (significance of the overlap). Additionally, if the regression analysis is the gold standard, why not use the regression instead of BAE?

Was there anything done to independently confirm the subtypes of Sst-expressing neurons in the subgroup analysis correspond to a biological process and not data artifacts?

5- Power Analysis:

The claim about the power of clustering in small groups (“Even if a very small group is captured by such a clustering, the statistical power to infer marker genes is often limited due to the small sample size”) could benefit from analysis that demonstrates the proposed method provides higher power compared to alternatives.

Recommendation:

The authors address several challenges in this paper. While it is commendable, it results in the manuscripts message becoming somewhat vague. A more focused approach will improve clarity and highlight the main contributions of BAE. Focusing on a limited number of claims providing robust empirical evidence will achieve this. The most exciting aspects of this paper are scenarios where the BAE pipeline excels, such as the time-series analysis for the development cycle, and specification of subtypes.

Minor Issues:

1. Terminology:

The authors use the expression “dimension reduction” which is technically not wrong. However, “dimensionality reduction” is the more commonly used term and is recommended.

2. Methods section:

The methods section is thorough and considers the audience needs by going into details. However, clarity could be improved. There might be a mistake on page 18, section 5.2, first paragraph, in the definition of dimensionality of $\mathbf{W}^{(l)}$. It seems n_i and n_{i-1} should be swapped. Or notation should be updated later to indicate that matrix is transposed.